# A highly utilized and practical lithium-sulfur positive electrode enabled in all-solid-state batteries

Ashley Cronk[1], Xiaowei Wang [2,5], Jin An Sam Oh [2], So-Yeon Ham [1], Shuang Bai[1], Phillip Ridley[2], Mehdi Chouchane[3], Chen-Jui Huang [3], Diyi Cheng[2], Grayson Deysher[1], Hedi Yang[3], Baharak Sayahpour[1], Marta Vicencio[2], Choonghyeon Lee[4], Dongchan Lee[4], Min-Sang Song[4], Jihyun Jang [2,6], Jeong Beom Lee [4] ✉ & Ying Shirley Meng [2,3] ✉

All-solid-state batteries using sulfur-based positive electrodes (cathodes) offer a cost-effective route to achieve high specific energy. However, low active material utilization and cycle life hinder performance. Here, we demonstrate a positive electrode design that employs sulfide solid-state electrolytes, where a high energy synthesis approach forms a metastable and ionically conductive interphase on the active material surface. This interphase facilitates high active material utilization and contributes capacity with cycling. We also show that tailoring active material particle sizes to the micron-scale improves rate performance and cycling stability. Structural analysis reveals that the substantial volume change of sulfur-based positive electrodes during operation can partially offset that of the negative electrodes, thereby mitigating internal mechanical stress. The combined design principles enable sulfur areal capacities up to 11 mAh cm$^{-2}$ while maintaining stable cycling at 25 °C. We further demonstrate several specific-energy-focused cell architectures, particularly a Li$_2$S anode-free pouch cell that operates under "low stack pressure" of 10 MPa. This work outlines practical design strategies for constructing high-specific-energy all-solid-state batteries for a broad range of emerging applications.

In the last decade, the need for safe and cost-effective energy storage systems has grown significantly. By 2030, the global demand for lithium-ion batteries is projected to double from 2.8 to 6 TWh[1], exceeding the projected supply. Due to the increasing adoption of electric vehicles and electrified aviation, much of this demand is driven by the transportation sector. While lithium-ion batteries using insertion-type positive electrodes have made substantial progress in terms of cost and energy density, these electrodes are reaching

capacity and performance limitations, necessitating the development of alternatives that are safer, lightweight, with lower cost to further advance electrification technologies[2,3]. All-solid-state batteries (ASSBs) using conversion-type positive electrodes, such as lithium-sulfur (Li-S), can overcome the shortcomings of current lithium-ion battery technology. Sulfur's high specific capacity (1675 mAh g$^{-1}$)[4] and abundance[5] make it a promising low-cost energy storage solution. ASSB architecture also improves operational safety by using non-flammable solid-

[1]Materials Science and Engineering Program, University of California San Diego, La Jolla, CA, USA. [2]Department of NanoEngineering, University of California San Diego, La Jolla, CA, USA. [3]Pritzker School of Molecular Engineering, The University of Chicago, Chicago, IL, USA. [4]LG Energy Solution, Ltd., LG Science Park, Seoul, Korea. [5]Present address: Chemical Sciences and Engineering Division, Argonne National Laboratory, Argonne, IL, USA. [6]Present address: Department of Chemistry, Sogang University, Seoul, Republic of South Korea. ✉e-mail: jaybilee@lgensol.com; shirleymeng@uchicago.edu

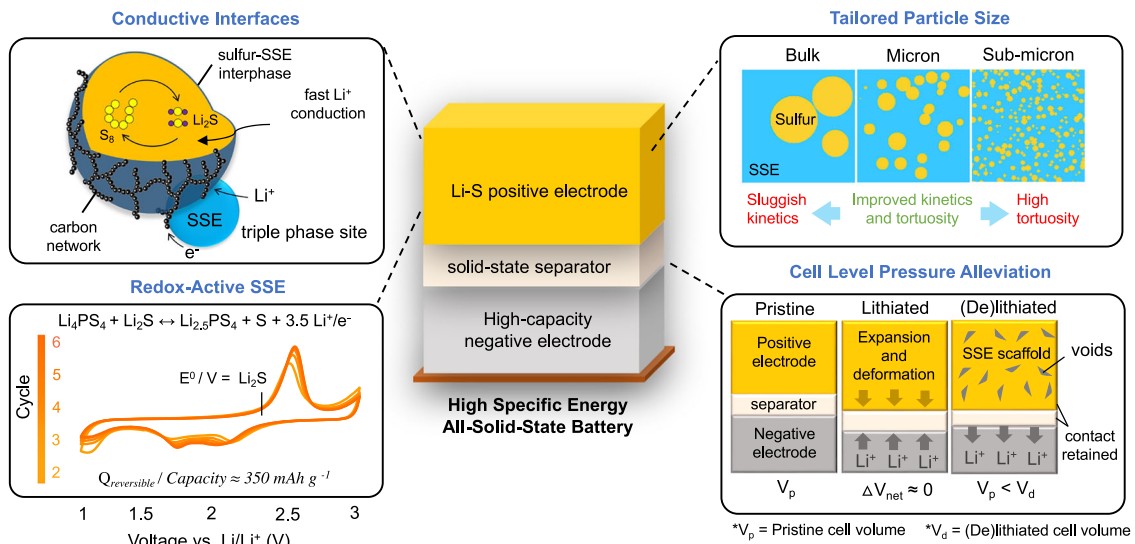

**Fig. 1 | All-solid-state battery using engineered Li-S conversion positive electrodes.** Key features required to enable high specific energy ASSBs, improving both utilization and cycle life under practical operating conditions.

state electrolytes (SSE) and eliminates the polysulfide dissolution and shuttling effect, a major challenge hindering the commercialization of Li-S in liquid electrolytes[6,7].

While ASSBs eliminate the polysulfide dissolution and shuttling effect, additional interfacial challenges inherent of SSEs are introduced. Efforts have shifted to addressing the slow kinetics and insulating properties of sulfur and $Li_2S$, in addition to their chemomechanical degradation from expansion and contraction[8]. The conversion from $S_8$ to $Li_2S$ result in a 79% volume change[9] – ten times greater than that of conventional positive electrodes. This significant volume change generates high internal pressure on the surrounding SSE matrix, leading to void formation and poor interfacial contact[10]. Consequently, most Li-S solid-state studies have adapted electrode fabrication methods from liquid systems. Typically, active materials are incorporated within high surface area carbon hosts through ball-milling[11–14], solution processes[15–17], or vapor deposition[18,19] as a means to increase conductive interfaces and constrain volume changes. Nevertheless, these methods have resulted in inconsistent utilization and cycle life. Given the large amount of carbon typically used, inadequate ionic networks and insufficient contact to sustain conversion are likely responsible. Strategies like heat treatments[20] or creating 3D solid-electrolyte structures[21] demonstrate proof-of-concepts to enhance interfacial contact. Incorporating catalysts has been found to overcome poor contact by improving conversion kinetics[22,23], although using critical elements like cobalt compromises the low-cost novelty of the Li-S system. While some of these approaches have shown improvement compared to the liquid system, high areal loadings with long cycle life necessary for practicality, have yet to be demonstrated.

Li-S conversion requires "triple-phase" contact between the active materials, ionic and electronic network[8,24]. This is easier to achieve in liquid systems, as liquid electrolyte can flow through electrode pores. In ASSBs, intimate solid-solid contact can be limited and dependent on electrode architecture, including optimal particle sizes and their distribution. Ideally, a uniform distribution of active materials, SSE, and carbon should be achieved. Common synthesis approaches to achieve this can occasionally promote decomposition or redox activity when using sulfide-based SSEs[16,25,26], formally associated with generating irreversible decomposition products at high voltages[27], especially with high surface area carbon[28]. However, the lower operating voltage of sulfur conversion is more compatible with the lower oxidation stability of sulfide-based electrolytes, where incomplete redox can be reversible[29,30]. Leveraging the redox activity of sulfide SSEs can

enhance the reaction kinetics of sulfur and $Li_2S$[31] and ultimately improve cycling stability.

While SSE redox can potentially enhance reaction kinetics and cycle life, sustaining interfacial contact between Li-S active materials and SSE remains a challenge. Interfacial reactions between sulfur or $Li_2S$ and sulfide SSEs can result in the formation of ionically conductive phases. Previous studies by Lin et al., found that $Li_3PS_4$ (LPS) and sulfur can react to form a sulfur rich thiophosphate phase ($Li_3PS_{4+n}$)[32], creating bridged sulfur chains no longer in cyclo-$S_8$ form. Similar strategies have also been carried out with $Li_6PS_5Cl$ (LPSCl) and sulfur[15] for its higher ionic conductivity. However, previous synthesis routes employed solvents to facilitate these reactions, which can introduce complexities to the fabrication process. Ideally, a solvent-free synthesis could be employed. Tanibata et al., showed that sulfur can bond to $P_2S_{5+n}$ units using a mechanochemical milling process, which delivered a high initial discharge capacity[33]. Leveraging both SSE redox activity and forming ionically conductive interphases can potentially solve the utilization and stability challenges still observed with Li-S positive electrodes. While mechanochemical synthesis has been widely used, standards have yet to be established, likely due to the many synthesis permutations that can be employed. Therefore, the impact of mechanochemical synthesis on Li-S electrode architecture is still not fully understood.

Here, we present a Li-S positive electrode design that overcomes interfacial, kinetic, and chemomechanical challenges when implemented in ASSBs (Fig. 1). A high energy single step short duration mechanochemical process creates ionically conductive interphases on the sulfur particle surface through sulfur bonding between the solid electrolyte and sulfur particle. To enhance cycle life, three features were enabled. First, activation of the sulfide SSE redox activity was accomplished using the synthesis process and was confirmed to be reversible within the sulfur and $Li_2S$ voltage windows. Using differential capacity analysis and X-ray absorption spectroscopy (XAS), we deconvolute these capacity contributions for both sulfur and $Li_2S$ positive electrodes. Second, tailoring the active material particle sizes to the micron scale reduces ionic tortuosity, enabling high-rate cycling. Third, structural analysis confirms that both sulfur and $Li_2S$ positive electrodes undergo significant volume change. We also reveal the mechanical disadvantage of $Li_2S$ compared to sulfur due to its volume contraction behavior. Nevertheless, the volume change of the positive electrodes was found to balance the volume change of negative electrodes, alleviating internal pressures within the cell, especially

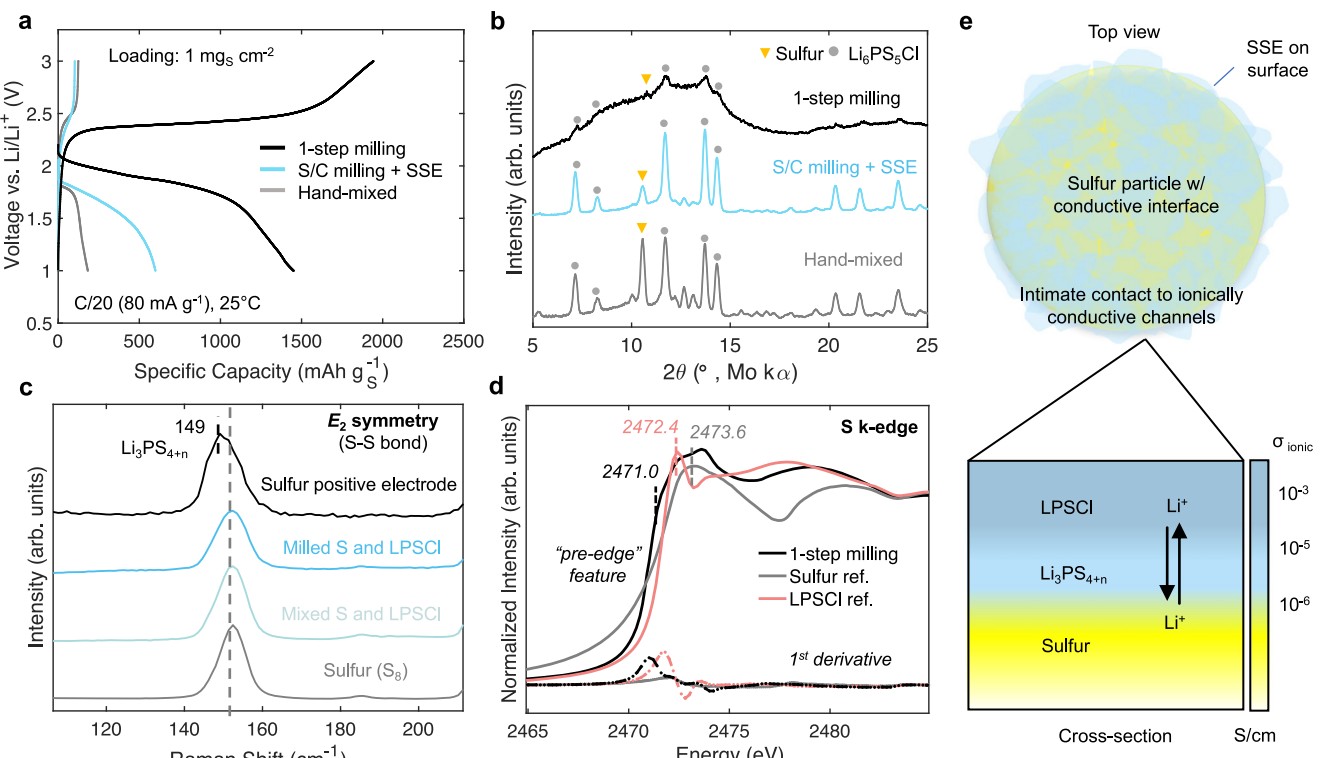

**Fig. 2 | Characterizing sulfur positive electrodes after synthesis. a** First cycle voltage profiles of the investigated positive electrode preparation methods in Li-In half cells operating with a rate of 80 mA g$^{-1}$. Cells were evaluated at room temperature: 25 °C ± 1 °C. **b** X-ray diffraction (XRD) spectra of the prepared sulfur positive electrodes. **c** Raman spectroscopy and (**d**) X-ray absorption spectroscopy (XAS) spectra and corresponding 1$^{st}$ derivates of the positive electrode fabricated using the one-step milling process. Intensities are normalized to compare relative features, especially at the pre-edge. **e** Schematic illustrating the surface reaction between sulfur and LPSCl, where a sulfur-rich ionically conductive interlayer is formed on the sulfur particle surface after the 1-step milling process.

when paired with high-capacity negative electrodes such as silicon. As a result, stable cycling was achieved with various solid-state cell configurations utilizing both sulfur and Li$_2$S positive electrodes. Sulfur positive electrodes at 11 mAh cm$^{-2}$ delivered stable cycling performance with 87 % retention for over 140 cycles at 25 °C. A 'proof of concept' Li$_2$S||anode-free solid-state pouch cell delivers a high reversible capacity of 900 mAh g$^{-1}$ under practical operating conditions, without catalysts or liquid electrolytes. This study provides an in-depth investigation into Li-S positive electrode and cell level design, supporting the development of practical and high specific energy ASSBs using Li-S conversion chemistry.

## Results and discussion
### Creating conductive interfaces
Positive electrode architecture including conductive interfaces are one of the critical components to enable high performing Li-S ASSBs. The reactivity between the active materials and the sulfide SSE was investigated using a one-step mechanochemical milling process, where high milling intensities were used for a short duration. Milling intensity influences the reaction kinetics by tuning the collision frequency between particles. High intensities have been found to accelerate reaction products often leading to shorter process times[34,35]. These two parameters were found to be critical in creating conductive interfaces between sulfur and the sulfide SSE without degrading ion conduction pathways within the composite. To prove the effectiveness of this approach, two common fabrication methods were re-visited: hand-mixing and a multi-step milling process where the sulfur, carbon, and SSE are milled separately. Schematics illustrating each method and their expected distribution are shown in Fig. S1. A commonly used sulfide electrolyte, carbon type, and weight composition was

employed for our study, where unmodified elemental sulfur, argyrodite SSE (LPSCl), and acetylene black (AB) carbon were used to fabricate the positive electrode with a weight ratio of 30:50:20 wt%. This composition was used in all electrochemical tests unless otherwise stated. Prior to positive electrode composite synthesis, LPSCl was premilled to the micron scale to induce more interfacial contact to sulfur (Fig. S2).

To compare the different positive electrode synthesis methods, cells were fabricated with LPSCl as the separator layer and a Li$_1$In alloy as the negative electrode. Figure 2a shows their first cycle voltage profiles where increases in utilization and capacity are attained by introducing high energy milling steps to the fabrication process. The multi-step process however suffered from low utilization, implying insufficient sulfur-SSE contact. It was also irreversible, marked by a low conversion efficiency (CE) of 17%. Commonly observed in prior works[13,18,25,36,37], a low CE suggests insufficient mass transport to reconvert Li$_2$S back to sulfur, usually requiring high activation potentials to do so[38]. The low CE can also be from the isolation of sulfur particles and active surface areas after volume expansion. A comparison between hand-mixing and ball-milling has been previously investigated by Ohno et al., where ball-milling achieved higher capacities, resulting in a CE of 100%[39]. However, the single-step method used here delivered a discharge capacity near the theoretical (~1500 mAh g$_S^{-1}$ at 25 °C) and a CE of 129%. This means additional capacity from the SSE has been obtained. A high discharge capacity coupled with additional capacity from the SSE, suggests that this fabrication strategy produced an architecture to facilitate improved ionic/electronic transport and "triple phase" sites for conversion. Mechanochemical synthesis has been a common method to incorporate solid state materials. Usually, this is done using low milling intensities, long milling times, and in

multiple steps to mitigate SSE decomposition, recreated above. The one-step approach introduced here subjects all components to the high energy milling process, likely altering the physiochemical properties of sulfur and LPSCl.

The diffraction patterns of the positive electrode composites are shown in Fig. 2b. With all three methods, LPSCl and sulfur are detectable. Only the one-step method facilitated amorphization. Since the diffuse scattering of AB carbon can mask diffraction peaks, the milling procedure was done without carbon (i.e., S and LPSCl) (Fig. S4a), where the full width at half maximum (FWHM) increases for both characteristic peaks of sulfur and LPSCl. Amorphization, evidenced by peak broadening can be explained from the high energy milling process, which introduces heat, breaking down sulfur rings, and has been found to distort $PS_4^{3-}$ units, creating loose P-S bonds[40]. This is advantageous since crystalline sulfur (cyclo-$S_8$) requires large activation energies to break covalent bonds between sulfur atoms[41], lowering the energy barrier for conversion. Thermogravimetric analysis (TGA) was used to quantify elemental sulfur within the positive electrode composite (Fig. S5), which should sublime at 350 °C[42]. However, 6.5 wt% of sulfur is unaccounted for, indicating an alteration of the sulfur bonding environments and possible reaction with LPSCl. To verify this, Raman spectroscopy was conducted focusing on the S-S bending region centered at 152 cm$^{-1}$, a signature peak of $E_2$ symmetrical bending[43]. In this region, a redshift is observed with the milled S and LPSCl and one-step milled composite which includes carbon (Fig. 2c). The S-S bending at lower wavelengths suggests an increase in bond lengths from P-S stretching, confirming the formation of sulfur rich thiophosphates ($Li_3PS_{4+n}$), where elemental sulfur bonds with the sulfur at the $PS_4^{3-}$ terminals of LPSCl[15,32]. This observation is complimented by both a reduction in intensity and wavenumber for the symmetric P-S stretching of the thiophosphate unit ($PS_4^{3-}$) in LPSCl at 425 cm$^{-1}$ (Fig. S4b), suggesting polymerization with bridging S-S bonds[44]. The bonding between the thiophosphate units of LPSCl and sulfur, facilitated by the high energy synthesis used in this work, ensures intimate contact of the sulfur and SSE interface, leading to high sulfur utilization shown in Fig. 2a. A high milling intensity is required to induce the interfacial reaction between sulfur and LPSCl. Sulfur utilization was found to decrease when using lower milling intensities (Fig. S3), suggesting incomplete formation of the $Li_3PS_{4+n}$ phase.

The X-ray absorption near-edge structure (XANES) region is useful in examining oxidation states and local bonding environments of sulfur. Synchrotron radiation at the Sulfur K-edge was measured for the sulfur positive electrode and reference samples (Fig. 2d). Two main peaks can be assigned to elemental sulfur at 2473.6 eV and LPSCl at 2472.4 eV. A pre-edge feature at 2470.1 eV is also evident from 1$^{st}$ derivative of the spectra. Pre-edge features indicate a reduction of the sulfur oxidation state and have been observed for long chain polysulfides ($Li_2S_y$)[45]. The pre-edge observed here is likely from chains of sulfur in $Li_3PS_{4+n}$. In addition to milling intensity, milling times are another important synthesis parameter. The ionic conductivity of composites consisting of only sulfur and LPSCl under high milling intensities (500 rpm) and increasing milling durations were measured, where an increase from $6 \times 10^{-6}$ S cm$^{-1}$ to $2 \times 10^{-5}$ S cm$^{-1}$ is observed after 1 h (Fig. S4c). The increase in ionic conductivity after 1 h of milling supports the formation of the ionically conductive sulfur rich phase. Standalone XRD, Raman, and EIS was measured for the LPSCl catholyte at 1- and 10-hour milling intervals (Fig. S4d–f). Contrary to what was observed for the S and LPSCl composites, continued milling of LPSCl alone decreases its ionic conductivity from $1.3 \times 10^{-3}$ S cm$^{-1}$ to $4 \times 10^{-5}$ S cm$^{-1}$, attributed to the formation of insulative $Li_2S$ and LPS phases. These results highlight the importance of limiting milling durations, where high milling intensities used for short durations were found to be sufficient to complete the interfacial reaction between sulfur and

LPSCl while preserving the ionic conductivity of the SSE and positive electrode.

Low-dose cryo-TEM, high-angle annular dark-field scanning transmission electron microscopy (HAADF-STEM) imaging, and elemental mapping was conducted on multiple particles of the positive electrode composite (Figs. S6 and S7). Hyperspectral mapping shows a homogenous distribution of all components. Line scan results show only Cl signal on the particle edges, attributed to LPSCl. At 50 nm from the particle edge, the Cl signal vanishes while only sulfur is present, suggesting that the bulk of the particle is sulfur. As the scan moves toward the center, the sulfur atomic fraction stabilizes to double the expected stoichiometry for LPSCl, revealing the sulfur-rich phase on the surface (Table S1). These results corroborate the interfacial reaction between sulfur and LPSCl, formed on the particle surface by the high energy synthesis process. This sulfur rich interphase lowers the energy barrier for lithiation, facilitating fast lithium transport from the SSE matrix into the sulfur bulk as illustrated in Fig. 2e. Sulfur can also bond to the thiophosphate units in $Li_3PS_4$ (LPS). However, the lower ionic conductivity of LPS compared to LPSCl (0.04 mS cm$^{-1}$ vs. 2 mS cm$^{-1}$) shown in Fig. S8a, will likely result in lower sulfur utilization when used as a catholyte. Cells were fabricated using LPS, where lower sulfur utilization was obtained (Fig. S8b). Although, reasonable capacity and cell performance are still achievable.

The interaction between LPSCl and $Li_2S$ was also investigated. A comparison between positive electrode preparation methods was conducted as done so for sulfur, where half cells were fabricated with a $Li_{0.5}In$ negative electrode. Comparable trends seen with sulfur are also observed with $Li_2S$ (Fig. S9). Hand-mixing the composite fails to cycle, and the multi-step process delivers low utilization coupled with large polarization. The one-step method, however, delivers a high specific capacity of 723 mAh g$^{-1}$ with a high coulombic efficiency of 99.3%, indicating good reversibility. With the $Li_2S$ cathode, the synthesis method resulted in the decomposition and amorphization of LPSCl, evidenced by undetectable peaks in the diffraction pattern and a shift of P-S stretching in $PS_4^{3-}$ from 425 cm$^{-1}$ to 418 cm$^{-1}$, assigned to $Li_3PS_4$ (LPS)[46] (Fig. S10a–b). This indicates that $Li_2S$ reduces LPSCl to LPS. Despite the catholyte decomposition and reduction in composite ionic conductivity (Fig. S10c), the $Li_2S$ cell exhibits stable cycling, where additional plateaus present in the voltage profile are attributed to SSE redox after the 1$^{st}$ cycle (Fig. S11). This confirms the redox activity from LPSCl decomposition products. Usually formed electrochemically, here the redox active products are formed during synthesis.

## Evaluating LPSCl redox activity with the influence of sulfur

The redox activity of sulfide SSEs has been responsible for delivering capacity beyond the theoretical when paired with Li-S positive electrodes[15,16,19] and associated with reversible[29] and irreversible[30] electrochemical behavior depending on the voltage cutoff and formed decomposition products. The impact of this behavior on Li-S ASSBs performance is still debated. Often overlooked in literature, deconvoluting the capacity contribution between sulfur and the solid electrolyte is essential in quantifying the effectiveness of the positive electrode architecture.

In this system, three active materials exist: bulk LPSCl, the formed interlayer phase ($Li_3PS_{4+n}$) consisting of sulfur that has reacted with LPSCl, and unreacted elemental sulfur. TGA was used to quantify the reacted and unreacted sulfur mass, where 23.5 out of the 30 wt% of elemental sulfur remains unaltered after synthesis. Using the known sulfur and LPSCl mass, we can estimate the capacity contributions of the active materials and compare to what was obtained experimentally. For a 1.6 mAh cm$^{-2}$ cell using a total composite mass of 2.5 mg, 23.5 wt% unreacted sulfur is expected to deliver 1 mAh (using a specific capacity of 1675 mAh g$^{-1}$). Therefore, during the discharge, unreacted bulk sulfur is estimated to contribute 83% of the capacity with the

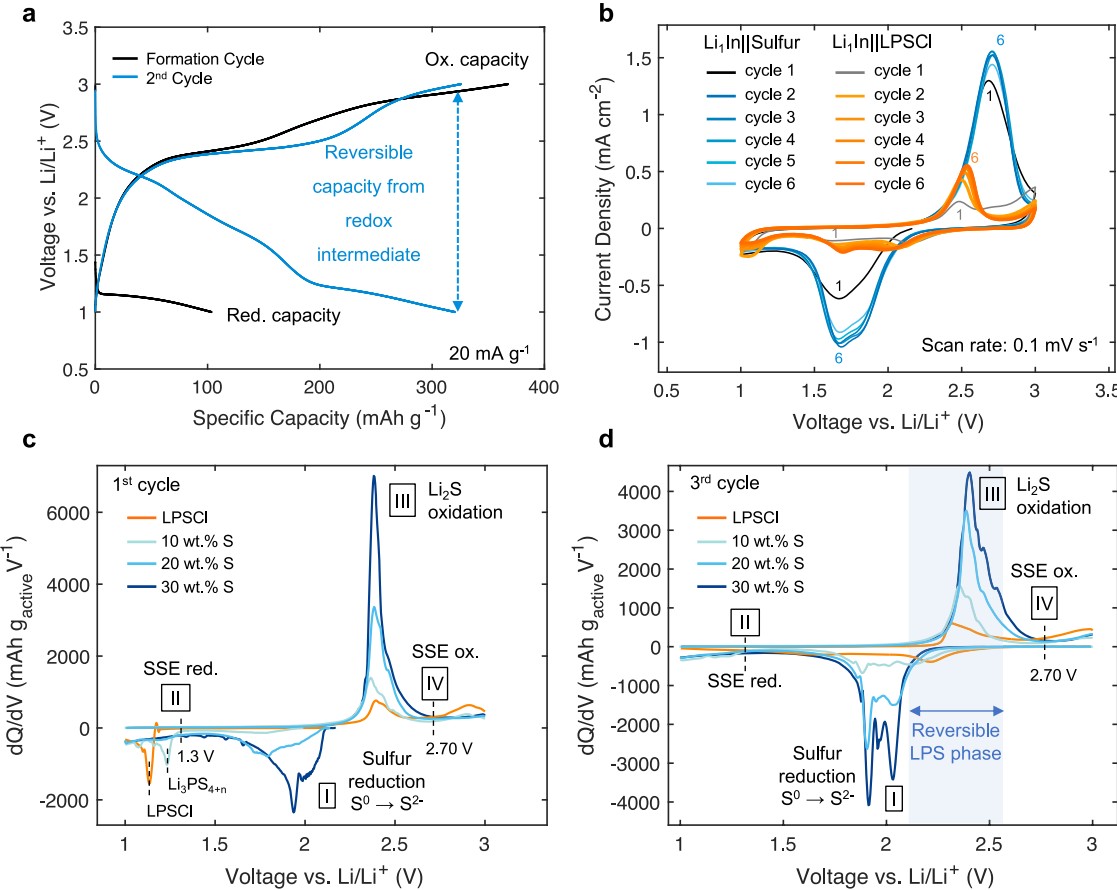

**Fig. 3 | LPSCl redox activity and capacity contribution.** Voltage profiles of Li-In || LPSCl half cells evaluated under (**a**) sulfur voltage limits. **b** Cyclic voltammetry of sulfur and LPSCl positive electrode composites with a Li-In negative electrode. The active mass of the sulfur positive electrode is 2.5 mg cm$^{-2}$, which consists of both sulfur and LPSCl. The active mass for the LPSCl positive electrode is 3 mg cm$^{-2}$. **c** dQ/dV curves of the 1$^{st}$ and (**d**) 3$^{rd}$ cycle for cells with decreasing sulfur wt% within the positive electrode composite. The areal loading of sulfur was 0.3 mg cm$^{-2}$, 0.84 mg cm$^{-2}$, and 1 mg cm$^{-2}$ corresponding to a total active mass of 2.8 mg cm$^{-2}$, 2.9 mg cm$^{-2}$, and 2.5 mg cm$^{-2}$ for 10 wt%, 20 wt%, and 30 wt% sulfur composites. Current densities used for dQ/dV were 60 mA g$_S^{-1}$, 70 mA g$_S^{-1}$, and 80 mA g$_S^{-1}$ for 10 wt%, 20 wt%, and 30 wt% sulfur composites, respectively. For all cells, the stack pressure was 75 MPa with a testing temperature of 25 °C ±1 °C.

remainder attributed to LPSCl and Li$_3$PS$_{4+n}$ reduction. To isolate the capacity expected from just LPSCl, cells with just LPSCl and carbon were fabricated using a Li-In alloy negative electrode as a lithium source. Constant current discharge and charge within the sulfur voltage limits delivered a reduction and oxidation capacity of 115 mAh g$^{-1}$ and 355 mAh g$^{-1}$ (Fig. 3a). Therefore, a much higher charge capacity from LPSCl is expected. Subtracting the expected LPSCl and unreacted bulk sulfur capacity from the total cell capacity obtained after the charge leaves the remaining capacity to be assigned to reacted sulfur in Li$_3$PS$_{4+n}$. This equates to a capacity contribution of 10% for the reacted sulfur and 27.5% for LPSCl after the charge, suggesting that the reacted sulfur in Li$_3$PS$_{4+n}$ and LPSCl are both redox-active and actively participates during cycling. This coupled redox behavior is observed between the sulfur positive electrode and LPSCl cells in the cyclic voltammetry (CV) scans (Fig. 3b).

The addition of sulfur will alter the redox behavior of LPSCl, especially after the high energy synthesis process. Additionally, the capacity contributions of LPSCl and sulfur in Li$_3$PS$_{4+n}$ during the discharge remains unknown. Therefore, cells were fabricated with lower sulfur weight percentages (10 wt% and 20 wt%) to highlight redox activity from the SSE and formed sulfur rich interlayer phase. Their capacities were normalized by their total active mass (i.e., sulfur and LPSCl) where differential capacity (dQ/dV) analysis was conducted to differentiate overlapping redox potentials and capacity contributions (Fig. 3c).

Four main regions are labeled in the dQ/dV curves. Region I highlights the dominant reaction during the discharge, attributed to the reduction of sulfur. Cells with higher sulfur wt% deliver most of their discharge capacity at this potential, meaning this is from unreacted bulk sulfur. Region II highlights the beginning of SSE reduction. Starting at 1.3 V, the reacted sulfur in Li$_3$PS$_{4+n}$ begins to reduce. This reduction peak is identifiable in the dQ/dV curves for all cells, but is mostly prominent for the cell with the lowest sulfur wt% due to a higher capacity contribution from this phase. The Li$_3$PS$_{4+n}$ phase is expected to follow similar redox as LPS, which reduces at a higher potential than LPSCl (Fig. S12). Therefore, we would expect the reduction of this phase to dominate until 1.15 V, where LPSCl begins to reduce. We can estimate their individual discharge capacity contributions based on the reduction potentials obtained from dQ/dV. From OCV to 1.3 V, 1 mAh is obtained, as estimated above. The remaining capacity from 1.3 to 1.15 V can be attributed to Li$_3$PS$_{4+n}$, which is responsible for 7.5% of the overall discharge capacity. The remaining capacity obtained from 1.15 to the lower voltage cutoff of 1 V is from LPSCl, which contributes 9.2%. The specific capacity of the reacted sulfur is estimated to be 553 mAh g$^{-1}$ during the discharge, likely existing in the Li$_3$PS$_{4+3}$ phase after synthesis. These calculations are presented in Table S2. The additional sulfur in the PS$_4^{3-}$ framework increases its functional capacity and reduces the electrochemical stability of the SSE. Region III is attributed to Li$_2$S oxidation. Higher wt% of sulfur will contribute higher capacity at this potential since more Li$_2$S

was formed during the discharge and is now being oxidized. The cell with just LPSCl and carbon will contribute capacity at this potential, attributed to LPS oxidation as investigated from previous work by Tan et al.[26]. Region IV starts at 2.7 V where SSE oxidation occurs. This additional oxidation capacity is recoverable upon the subsequent discharge, supporting the formation of a reversible redox-active LPS like phase. This is highlighted in the dQ/dV curves for the 3$^{rd}$ cycle (Fig. 3d), where reversible redox occurs at similar potentials as sulfur conversion and extends the obtained capacity at the nominal voltage. Despite a portion of sulfur reacting with LPSCl, the reacted sulfur is still utilized and can be estimated. Positive electrodes with 30 wt% sulfur is 84% utilized during the 1$^{st}$ discharge. Sulfur utilization significantly increases for lower sulfur weight percentages, increasing to above 90% for the positive electrode with 10 wt% sulfur. These estimations are listed in Table S3.

From the CV scans and dQ/dV curves, LPSCl and sulfur positive electrodes exhibit reversible electrochemical behavior. Reversible electrochemical behavior is also observed while operating within the $Li_2S$ voltage limits (Fig. S13). Assuming this behavior is consistent when paired with Li-S positive electrodes, redox products of the LPSCl electrolyte can be predicted. Complete oxidation of LPSCl forms sulfur, LiCl, and $P_2S_5$ with reduction products being $Li_2S$, LiCl, and $Li_3P$[26,29]. In both redox pathways, LPS is formed intermediately. Using the reduction and oxidation capacity of the LPSCl cells, the expected redox products can be hypothesized where upon discharge and charge, a lithium rich $Li_4PS_4$ and lithium deficient $Li_{2.5}PS_4$ can be formed. This is consistent with previous reports discussing the reversibility of the LPS phase[30,47].

$$Li_6PS_5Cl + 1\,Li^+/e^- \longrightarrow Li_2S + Li_4PS_4 + LiCl$$
$$Q_{red}/Capacity \approx 100\,mAh\,g^{-1}$$

$$Li_2S + Li_4PS_4 + LiCl \longrightarrow Li_{2.5}PS_4 + S + 3.5\,Li^+/e^- + LiCl$$
$$Q_{ox}/Capacity \approx 350\,mAh\,g^{-1}$$

$$Li_2S + Li_4PS_4 \longleftrightarrow Li_{2.5}PS_4 + S + 3.5Li^+/e^-$$
$$Q_{reversible}/Capacity \approx 350\,mAh\,g^{-1}$$

It is expected that the formation of LiCl will reduce the effective ionic conductivity of the composite. However, its' formation does not seem to impact redox reversibility within this voltage window. However, the degree of LPSCl decomposition, like most metastable SSEs, is highly dependent on carbon contact and surface area, where a mixture of metastable ($Li_{6\pm x}PS_5Cl$) and stable phases likely exists in parallel[29]. LiCl interaction with other redox products formed within the cathode may help retain ionic conductivity and ionic conduction pathways. The upper voltage cutoff below 4 V limits the complete oxidation of LPSCl, preserving ionic conductivity with continued cycles. Meanwhile, the high oxidative tendency of LPSCl at 2.3 V vs. Li/Li$^+$ is also effective at reducing the activation potential of $Li_2S$ oxidation. In this work, the activation potential was found to be 2.4 V, without requiring the use of catalysts or kinetic promoters beyond the SSE itself.

To investigate the reversibility of the sulfur positive electrodes, redox products during the 1$^{st}$ formation cycle were probed. In-situ electrochemical impedance spectroscopy (EIS) was performed on the sulfur positive electrode at each state of charge (Fig. 4a). After discharge, an impedance growth of 85.2 Ω is observed, attributed to charge transfer resistance from $Li_2S$ formation. After charging, the charge transfer resistance reduces and returns near the pristine state, indicating good electrochemical reversibility (Fig. S15). Post-mortem analysis was conducted to validate $Li_2S$ formation and conversion, where diffraction peaks attributed to nanocrystalline $Li_2S$ are detected in the XRD spectra of the discharged sample (Fig. 4b), reinforcing the

high utilization of sulfur. We also observe that LPSCl regains crystallinity during electrochemical $Li_2S$ formation as compared to the pristine sulfur positive electrode. In the charged sample, $Li_2S$ is undetectable, indicating its complete oxidation, while the formation of sulfur at 10.5° 2θ is observed. The high charge capacity yet low peak intensity and amorphous background indicate that the reformed sulfur is amorphous as in the pristine state and suggests the sulfur rich interlayer is preserved with cycling. This was confirmed by conducting Raman spectroscopy on the cycled sulfur positive electrode, where a redshift to lower wavenumbers at the S-S bending region is observed (Figure S14), as previously shown for the pristine composite. A new diffraction peak at 15° 2θ is also identified after the 1$^{st}$ cycle. Raman spectra collected at a region within the SSE bulk shows a convoluted $PS_4^{3-}$ peak, attributed to a mixture of LPSCl and LPS (Fig. S16a and b). This new phase after the 1$^{st}$ cycle can be attributed to LPS, validating active participation and reversibility of LPS, as hypothesized from the electrochemical and dQ/dV results. Sulfur quantification using TGA also demonstrates good conversion efficiency (Fig. 4c). The mass loss differences between the pristine and charged composite originate from the sulfur rich interlayer, which increases the thermal stability of the reacted sulfur. The preservation of ionically conductive interfaces along with SSE redox behavior helps sustain sulfur utilization and cycle life.

Deconvoluting the redox products between LPSCl, sulfur, and $Li_2S$ is challenging due to their overlapping redox potentials. This can be challenging using XRD, which is unable to resolve amorphous phases, a prominent feature after electrode synthesis. Therefore, XANES spectra at the Sulfur K-edge was measured for Li-S positive electrodes at various states of charge. The reference spectra are reported in Fig. S17. Using the reference spectra, linear combination fit (LCF) analysis can be used to quantify species[48,49]. The fitted spectra for each electrode system, namely LPSCl (Fig. S18a–c), sulfur (Fig. S19a–d), and $Li_2S$ (Fig. S20a–c) positive electrodes, are reported in the supplementary information.

Figure 4d–f shows the weight distributions of the quantified redox products. For cells consisting of only LPSCl and carbon, half of LPSCl was found to decompose to LPS during the initial discharge, as hypothesized from the electrochemical results. After charging, 9.7 wt% of the composite is estimated to be sulfur with 22 wt% of LPS retained after the 1$^{st}$ cycle. LPS formation and retainment reinforces its role as a redox intermediate. The redox activity of LPSCl likely improves $Li_2S$ oxidation kinetics and contributes active mass from sulfur formation after charging. The LCF results for the sulfur positive electrodes are shown in Fig. 4e. After the initial discharge, 20.2 wt% of $Li_2S$ is estimated. This supports the high conversion efficiency of the positive electrode architecture. LPSCl also partially decomposes to LPS, as observed for the LPSCl system. A higher amount of LPS is also present than expected but is likely from the $Li_3PS_{4+n}$ phase from reacted sulfur. It must be noted that XANES quantification of the entire electrode can be challenging due to the shallow probe depth when using tender X-rays. Discussion regarding limitations of XANES quantification is included in the Supplementary Information. After charging, $Li_2S$ is no longer detected, with 32 wt% assigned to sulfur. The additional capacity observed after the formation cycle of the sulfur system can be attributed to the formation of LPS and sulfur from LPSCl oxidation. These results coupled with EIS, XRD, and TGA support the reversibility of the Li-S system, attributed to the formation of ionically conductive interphases and redox activity of the LPS phase from LPSCl decomposition. Similar conclusions can be drawn for the $Li_2S$ system. Complimenting findings revealed by XRD and Raman, half of LPSCl decomposes into LPS after synthesis (Fig.4f). Similarly for the sulfur system, the amorphous LPS/LPSCl mixture retains its ionic conductivity, supported by the $Li_2S$ electrochemical performance. A complete summary of the fitting results can be found in Table S5.

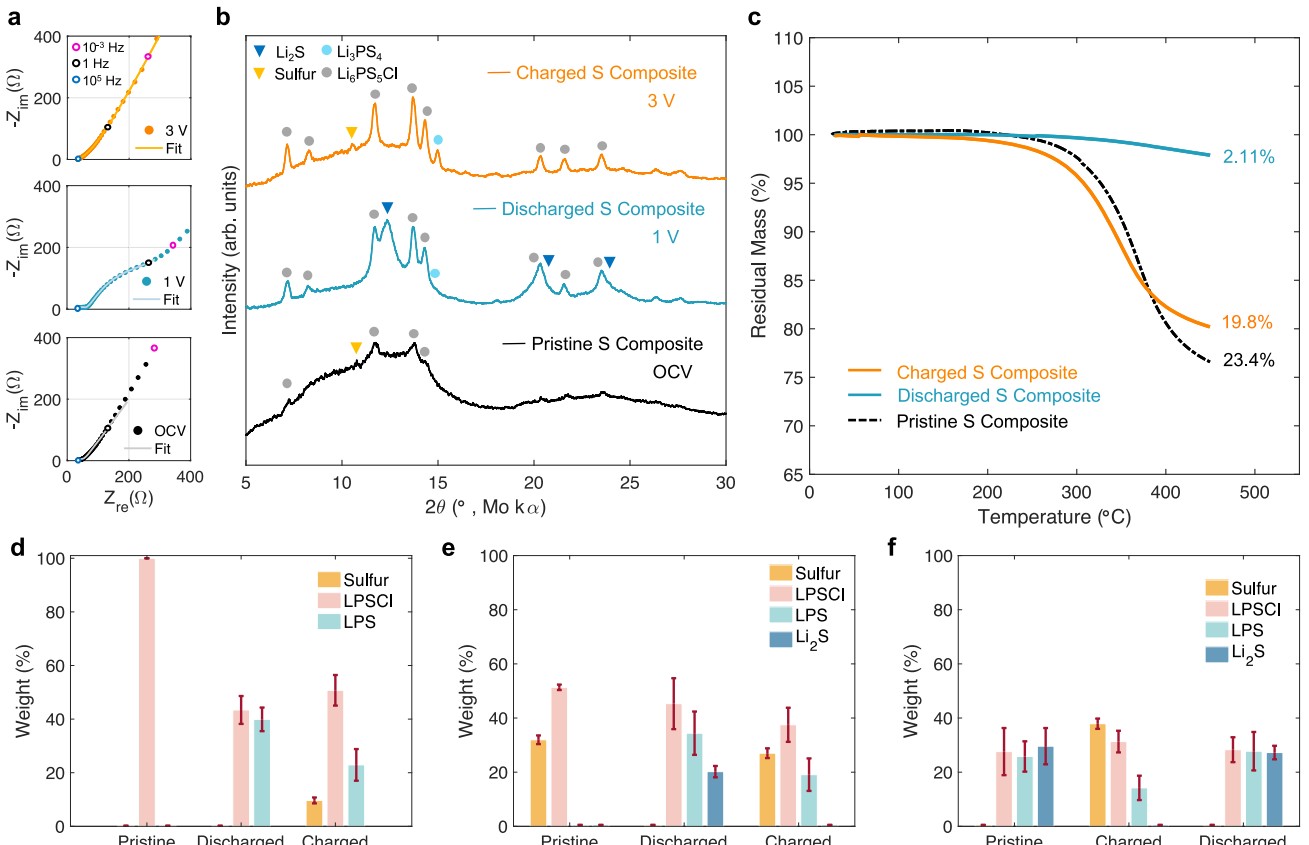

**Fig. 4 | Probing electrochemical reversibility. a** Nyquist plots of in-situ EIS measurements of cells with the sulfur positive electrode and Li-In negative electrode during the first formation cycle. **b** XRD and (**c**) TGA of high loading (7.6 mg cm⁻²) sulfur positive electrodes at the pristine, discharged, and charged states, performed in a nitrogen environment. Quantified sulfur masses are listed in

Table S4. XANES linear combination fitting results as weight percentages of products at each state of charge for (**d**) LPSCl (**e**) sulfur, and (**f**) Li₂S positive electrodes. Fit results are in Table S5. Error bars represent the standard error of the quantified masses. The electrochemical performance of cells used for XRD and XANES are presented in Fig. S21 and S22a-c.

## Enhancing rate and cycling stability

The previous electrochemical performances were obtained using as received sulfur and Li₂S. Their particle sizes were found to range drastically, with some particles on the order of 100 microns for sulfur (Fig. S23a–c) and 30 microns for Li₂S (Fig. S24a–d). After composite synthesis, which subjects sulfur to high energy milling with LPSCl and carbon, it is possible that particle sizes can reduce, although, sulfur particles were found to retain close to their original size as observed from the SEM-EDS mapping results (Fig. S25). These particle sizes, while sufficient for low-rate cycling, need to be reduced to improve contact and Li⁺ transport for higher rate operation. Sulfur particle sizes were produced on the micron and sub-micron scale (Fig. S26a–f). XRD was conducted after the size reduction process to ensure sulfur's crystallinity was preserved (Fig. S27). Given the many possible compositions, geometrical modeling was conducted to aid in experimental design, connecting sulfur particle size and mass fraction to active surface area and transport properties. Sulfur positive electrode geometries were stochastically generated with the sulfur as spherical particles and the carbon additives as aggregates (Fig. 5a). To compare with experimental capabilities, particle sizes for bulk, micron, and sub-micron sulfur ranged from 25 to 50 μm, 0.5 to 5 μm, and 0.25 to 0.5 μm, respectively. Figure 5b shows the evolution of the active surface area (the percentage of sulfur in contact with the SSE) as a function of sulfur content, represented as the active material (AM) mass fraction within the positive electrode. The reported values are the average over three repetitions.

Intuitively, the electrode with the smallest sulfur particles achieved the highest active surface area for all AM compositions, followed by the micron sulfur, and bulk sulfur electrodes. Therefore, sub-micron sulfur particles are expected to achieve the highest utilization due to more SSE contact. The tortuosity of the LPSCl phase was explored with the reported value being the average tortuosity in all directions. Surprisingly, the micron sulfur electrode exhibits the lowest ionic transport tortuosity, with all cases obtaining similar results until higher AM mass fractions (Fig. 5c). The geometrical modeling results suggest that sub-micron particles may be the best choice to achieve high utilization, yet high active wt% may be challenging to implement since the ionic transport tortuosity increases drastically after 50 wt%. Despite the expected high utilization with sub-micron particles from higher surface areas, it must be noted that the geometrical modeling does not consider the chemomechanical effects from lithiation.

To verify the modeling results, room temperature electrochemical performance was performed evaluating each sulfur particle class in Li-In half cells. All particle sizes deliver comparable discharge capacities during the 1ˢᵗ formation cycle at low rates of 80 mA g⁻¹ (C/20) (Fig. 5d). The sub-micron sulfur cell delivers the highest discharge capacity of 1694 mAh g⁻¹, beyond the theoretical. This was followed by micron and bulk sulfur which delivered 1615 mAh g⁻¹ and 1500 mAh g⁻¹, respectively. This means that more LPSCl redox activity can be activated with higher surface area particles. Higher charge capacities are also observed for all cells. The additional charge capacity results in a

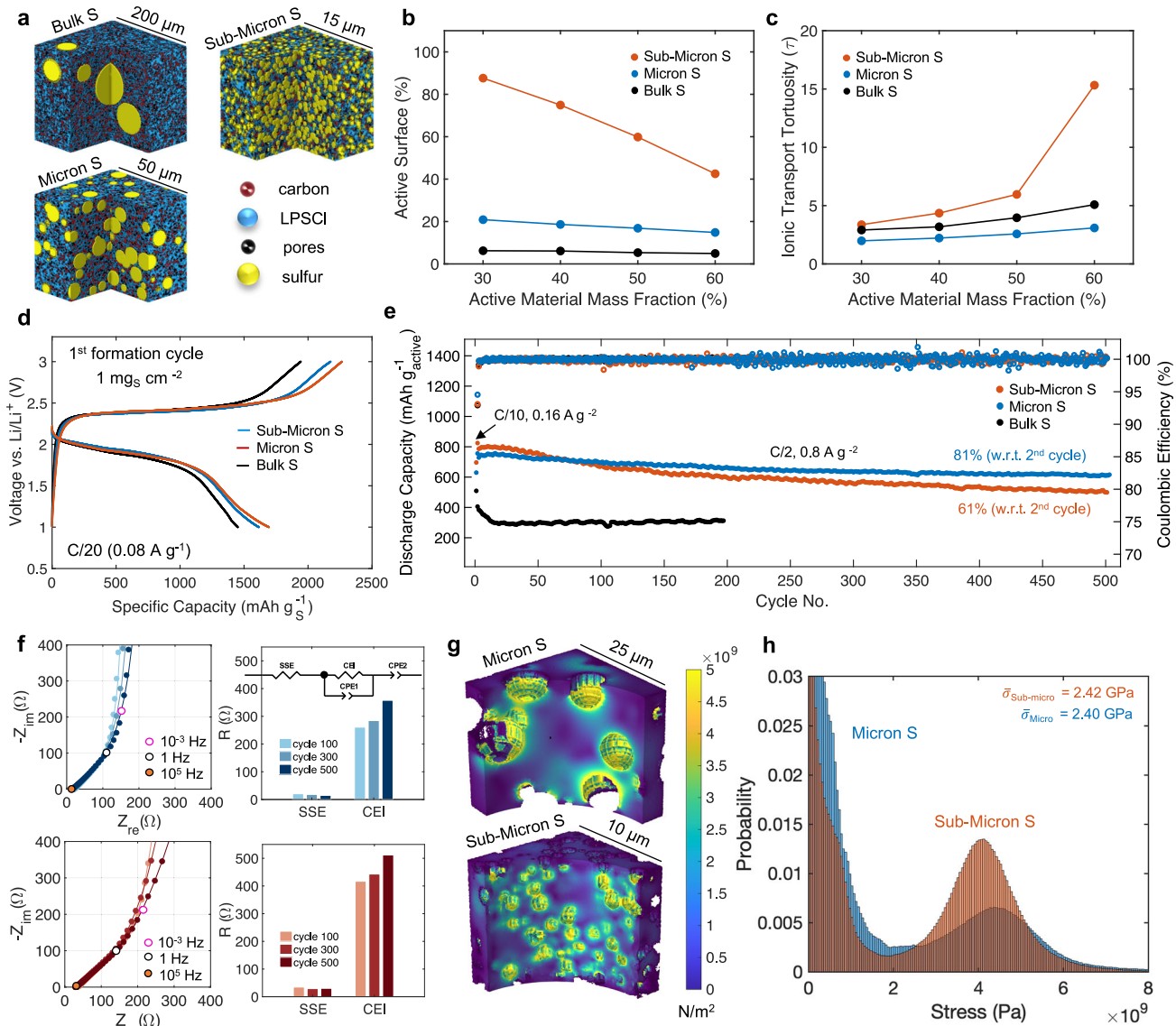

**Fig. 5 | Modeling sulfur positive electrodes microstructures with electrochemical validation in Li-In half cells. a** Geometrically modeled sulfur positive electrodes for bulk, micron, and sub-micron sulfur particles. **b** Specific active surface area and (**c**) ionic transport tortuosity as a function of AM content (%) and particle size. **d** First cycle voltage profiles of positive electrode composites with various particles sizes **e** Long-term cycling stability at C/2 (800 mA g⁻¹) with discharge capacity being normalized by the active mass (sulfur and LPSCl). **f** Nyquist plots and equivalent circuit fitting results from EIS measurements of the micron (top) and sub-micron (bottom) sulfur cathodes at cycle 100, 300, and 500 in their de-lithiated states. Cycling and EIS measurements were collected at 25 °C ± 1 °C. **g** Distribution of the von Mises stress on the SSE matrix after simulated lithiation. **h** Predicted variation of maximum von Mises stress. Data for each electrode and iteration is available in the figshare repository under accession code https://doi.org/10.6084/m9.figshare.31094524.

higher discharge capacity from the 2nd formation cycle with a reduction in cell polarization from 577 mV to 452 mV (Figure 28a). The reduced cell polarization is from the LPSCl and Li₃PS₄₊ₙ redox products and potentially beneficial chemomechanical behavior that improves contact during volume expansion. Rate capability was conducted up to 1 C (1.6 A g$_S^{-1}$ or 1.6 mA cm⁻²) (Fig. S28b). The current density at 1 C is beyond the critical current for the Li-In negative electrode (~1 mA cm⁻²)[50], however, all three cells deliver reasonable utilization at all rates, with bulk sulfur delivering the lowest due to kinetic limitations within the positive electrode. Since LPSCl was found to contribute capacity, the specific capacity considering both active masses (i.e., sulfur and LPSCl) show reasonable utilization (700 mAh g$_{active}^{-1}$) for both micron and sub-micron at these rates. This is critical for solid-state, as the catholyte is usually considered inactive, responsible for "dead weight" within the cell. In this work, the catholyte contributes electrochemically to the overall cell capacity.

Long term high-rate cycling was evaluated to investigate the particle size effect on cycle life (Fig. 5e). The bulk sulfur composite exhibits fast decay, due to longer Li⁺ transport lengths from large sulfur particles and poor contact observed after densification (Fig. S23c). Despite achieving the highest utilization, the sub-micron sulfur composite delivers 61% retention, with micron being the most stable system, achieving 81% retention after 500 cycles. A 20% increase in retention with the larger micron scale particles is a significant improvement. The capacity fade observed with the cell using the sub-micron sulfur particles may be due to excessive decomposition from higher surface area between the carbon and SSE. To test this, cells were constructed with the micron sulfur composite using carbons that possess different specific surface areas and morphologies (Fig. S29a–e), each chosen to intentionally facilitate more SSE redox activity. Increasing the carbon surface area increased utilization, mainly from the SSE, but showed no impact on cycling stability. This

means the decay observed with the sub-micron sulfur composite is likely of (chemo)mechanical origin, causing degradation at the positive electrode and SSE particle interfaces. This interfacial degradation should result in impedance growth. Therefore, EIS of the micron and sub-micron sulfur cells was measured after 100, 300, and 500 cycles (Fig. 5f). The intermediate frequency range was assigned to the interfaces within the positive electrode (cathode electrolyte interface)[51]. The sub-micron sulfur cells were found to possess a higher resistance after all cycles, attributed to more mechanical degradation with continued cycling. However, these results differ marginally (Fig. S30a, b). Cross-section scanning electron microscopy (SEM) imaging may be useful, although both electrodes were found to show indistinguishable morphology (Fig. S31a, b). Therefore, mechanical-based simulations using the generated electrodes in Fig. 5a may shed insight on the accumulated stresses at these interfaces.

Sulfur undergoes large volume expansion upon lithiation[52] resulting in internal stresses at its interfaces. To simulate this mechanism, volume expansion was estimated based on % utilization for each particle size using 1st cycle discharge capacities, subtracting the expected capacity from LPSCl (Fig. S32). Simulations were conducted using finite element method (FEM) where the sulfur particles underwent the prescribed volume expansion derived from the electrochemical results. Parameters and equations used in the simulations can be found in Tables S6 and S7. Figure 5g displays the distribution of maximum von Mises stress on the SSE matrix, where for both micron and sub-micron cases, stresses beyond 5 GPa were predicted. A bimodal stress distribution shown in Fig. 5h shows that most of the SSE matrix does not experience stress since only 30 wt% of sulfur is used in the simulation. However, at the sulfur particle and SSE interface, the sub-micron sulfur composite experiences a higher frequency of stress from more volume expansion and increased tortuosity within the electrode. Stress accumulated at the boundaries of the sulfur particles can propagate when these particles are in proximity, creating thinner SSE channels that experience much higher stress versus the bulk. If SSE fracture occurs, pore formation is possible after delithiation. These simulations only capture stress after the 1st lithiation. However, SSE degradation is expected to accumulate with cycling, disrupting ion conduction pathways and leading to more capacity fade as observed in Fig. 5e. Electrochemical evaluation, coupled with FEM simulations uncover that sub-micron sulfur particles create a microstructure with high electrode tortuosity, resulting in higher interfacial stresses and faster capacity decay. Nano-scale sulfur and Li₂S particles can be used to achieve high utilization[53–55]. However, in this work we find that micron-scale particles balance both utilization and stable cycling, critical for practical Li-S cathodes.

## Morphological evolution of Li-S positive electrodes

In ASSBs, Li-S positive electrodes are expected to experience chemomechanical degradation from their large volume changes, resulting in stress on particle interfaces as simulated above. Electrochemical performances of sulfur, Li₂S, and LPSCl were found to be highly reversible, but this impact on positive electrode morphology remains unknown. To visualize this, cross-sectional SEM images were prepared for both micron sulfur and Li₂S cells at each state of charge (following the same protocol in Fig. 4). Both pristine sulfur and Li₂S positive electrodes exhibit a dense structure, possessing a thickness of 26 μm and 45 μm, respectively, and good interfacial contact (Figs. 6a and 6b). Calculations were done to estimate the volume change (%) as a function of sulfur wt% assuming complete lithiation (Fig. S33). Here, 25.8 vol.% change is expected, resulting in a thickness increase to 32.7 μm, close to the observed result. The lithiation capacity of this cell was 1.22 mAh. Therefore, the thickness increase of the positive electrode equates to 4.9 μm mAh⁻¹, which is also close to the expected thickness growth of Li metal with cycling[56]. Therefore, lithiating the sulfur positive electrode can compensate the volume reduction from stripping Li metal.

After completing the 1st cycle, the sulfur positive electrode thickness was retained but can be resolved by an increase in porosity. These results reveal that during lithiation, the resulting stress on the SSE matrix causes it to plastically deform to accommodate particle expansion. The deformed structure preserves intimate contact to the separator layer and assists structurally during cycling. Subsequent volume expansion is likely supported by the pre-deformed SSE structure. The plastic deformation of the SSE matrix is not surprising, as the simulated stresses estimated above is near the shear modulus of LPSCl[57].

Li₂S is expected to shrink after delithiation. Shown in Fig. 6b, the Li₂S positive electrode thickness decreases drastically (~ 40%), where columnar cracking is observed in the cross-section as well as on the surface (Fig. S34a, b). Surface cracking is likely strain induced from inhomogeneous lithium removal. This morphological phenomenon has been observed in ASSBs using silicon negative electrodes[58] and these results suggest that this chemomechanical behavior occurs for conversion electrodes when constrained to 2D interfaces. After completing one cycle, the electrode morphology is however, reversible, returning close to the pristine state and is a similar thickness as the electrochemically formed Li₂S in Fig. 6a. These results provide two key insights. First, conversion positive electrodes can alleviate internal pressure from negative electrode volume changes. Second, Li₂S positive electrodes inherently face more mechanical challenges compared to sulfur due to strain induced cracking after delithiation.

To demonstrate the pressure alleviation proof of concept, operando pressure monitoring was conducted comparing LiCoO₂ (LCO), sulfur, and Li₂S when paired with pure μSi and lithiated Si negative electrodes. LCO will expand due to Jahn-Teller distortions[59], although volume expansion is 10 times less than Li-S positive electrodes. Si exhibits a high specific capacity and undergoes near 200% volume expansion upon lithiation, however lower than expected due to the constrained 2D interface in solid-state[58]. Nevertheless, pressure imbalance and resulting chemomechanical degradation is one of the most challenging aspects for solid-state silicon negative electrodes[60]. In theory, a mechanically balanced system can be achieved using conversion positive electrodes[61]. Since a lithium source is required for sulfur, cells were assembled using a lithiated silicon negative electrode, where minimal pressure changes with sulfur are observed (Fig. 6c). However, the LCO cell pressure increases five times that observed for the sulfur case, with pressure fluctuations consistent with previous studies[62]. The μSi||Li₂S cell shows almost zero pressure variation during the 1st formation cycle (Fig. 6d), partially responsible for the stable cycling performance shown in Fig. S35, with 83% retention after 140 cycles. These results demonstrate that pressure changes from high-capacity negative electrodes can be compensated using conversion positive electrodes that are highly utilized. This improves cycle life and mitigates cell 'breathing' during cycling, an important consideration for higher loading cells and when integrating cells into pack level architecture.

## High loading electrochemical evaluation

High loading electrodes are required to achieve high cell specific energy. Therefore, cells with increasing sulfur loadings were constructed and first paired with a LiIn alloy negative electrode (Fig. 7a). Areal capacities of up to 10 mAh cm⁻² with a discharge capacity of 1314 mAh g⁻¹ was obtained (Fig. 7b), although with a slight increase in polarization, attributed to the high sulfur loading of 7 mg cm⁻² evaluated at 25 °C. Due to the limited critical current density of LiIn alloys, cycling performance of this high loading system was evaluated at relatively low current densities of 0.52 mA cm⁻² (Fig. 7c), showing minimal decay, with 86.8% retention after 140 cycles. The ability to achieve stable cycling at 11 mAh cm⁻² at room temperature (25 °C), demonstrates the effectiveness of the positive electrode microstructure. To overcome the low current density of LiIn, free standing

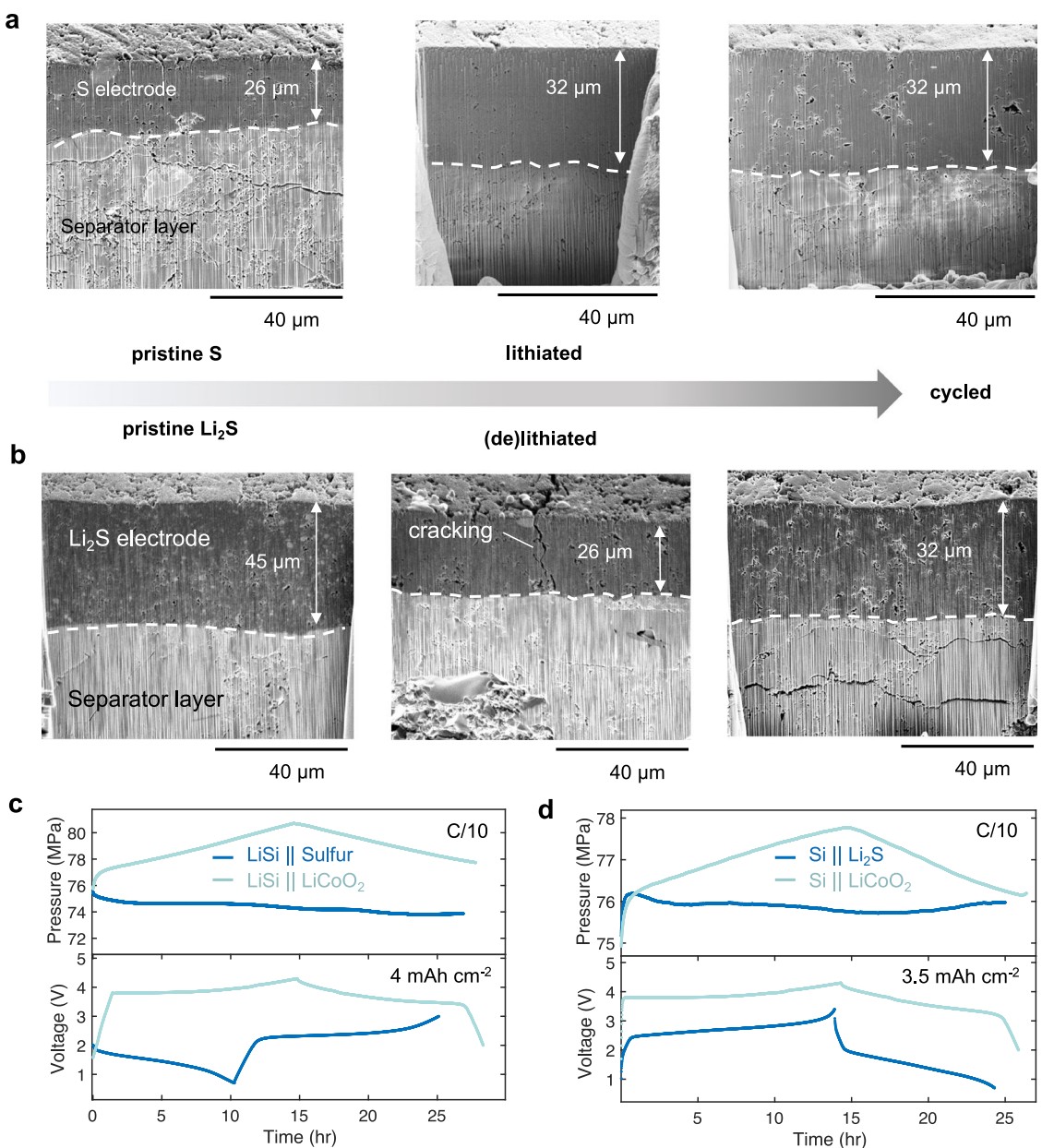

**Fig. 6 | Quantifying positive electrode and cell level volume changes.** Cryo-FIB images of the (**a**) micron sulfur and (**b**) Li$_2$S positive electrode composite at various states of charge. **c** *Operando* pressure monitoring and voltage profile of LCO and Sulfur paired with a lithiated silicon negative electrode and (**d**) LCO and Li$_2$S paired with a μSi negative electrode during the first formation cycle. Cells were cycled with a C-rate of C/10. LCO, Sulfur, and Li$_2$S mass loading was 26 mg cm$^{-2}$, 2.5 mg$_S$ cm$^{-2}$, and 3.8 mg cm$^{-2}$, corresponding to a specific current of 0.1 A g$^{-1}$, 0.16 A g$^{-1}$, and 0.10 A g-1, respectively. Testing temperature: 25 °C ± 1 °C. N/P ratio: 2.

dry process positive electrode films were fabricated and paired with a Li$_2$Si negative electrode (Fig. 7d). These cells were found to deliver reasonable capacities up to 1 C (5.5 mA cm$^{-2}$) as shown in Fig. 7e, with full recovery at C/20 (0.3 mA cm$^{-2}$). This dry process Li$_2$Si||Sulfur system also demonstrates stable cycling performance at reasonably high loadings 7.4 mAh cm$^{-2}$, resulting in 77.4% retention after 150 cycles at 1.5 mAh cm$^{-2}$ current densities (Fig. 7f).

Pouch cell form factors utilizing thick dry process positive electrodes, thin separator layers, and high-capacity negative electrodes are necessary to achieve high specific energies in all-solid-state[63]. In addition, high areal capacity and active weight percentages are also required. Illustrated in Fig. 7g, even lower weight percentages of sulfur at 10 mAh cm$^{-2}$ can realize 500 Wh kg$^{-1}$ and is likely a more promising approach than increasing the weight percentage of sulfur. This is because after lithiation, the volume of SSE compared to lithiated sulfur

will reduce. Despite sulfur possessing a higher specific capacity than its Li$_2$S counterpart, negative electrode selection is limited to those with a lithium source, making lithium metal or pre-lithiated silicon a common option, regardless of their high costs and manufacturing challenges. Using the positive electrode composition reported in this work, Li$_2$S can achieve over 400 Wh kg$^{-1}$ using silicon as the negative electrode and go beyond 500 Wh kg$^{-1}$ if combined with an anode-free architecture (Fig. 7h). To demonstrate the capabilities of the positive electrode architecture using Li$_2$S as the active material, a 15 mAh pouch cell was constructed without a negative electrode, utilizing dry process methods, and reducing the separator layer thickness from 500 μm to 50 μm (Fig. 7i). Industry standard formation rates and cycling protocols were used, where this configuration delivers high utilization (1077 mAh g$^{-1}$) and an initial coulombic efficiency of 83% (Fig. 7j). Stable cycling at C/3 (1.5 mA cm$^{-2}$ or 0.34 A g$^{-1}$) was also achieved at

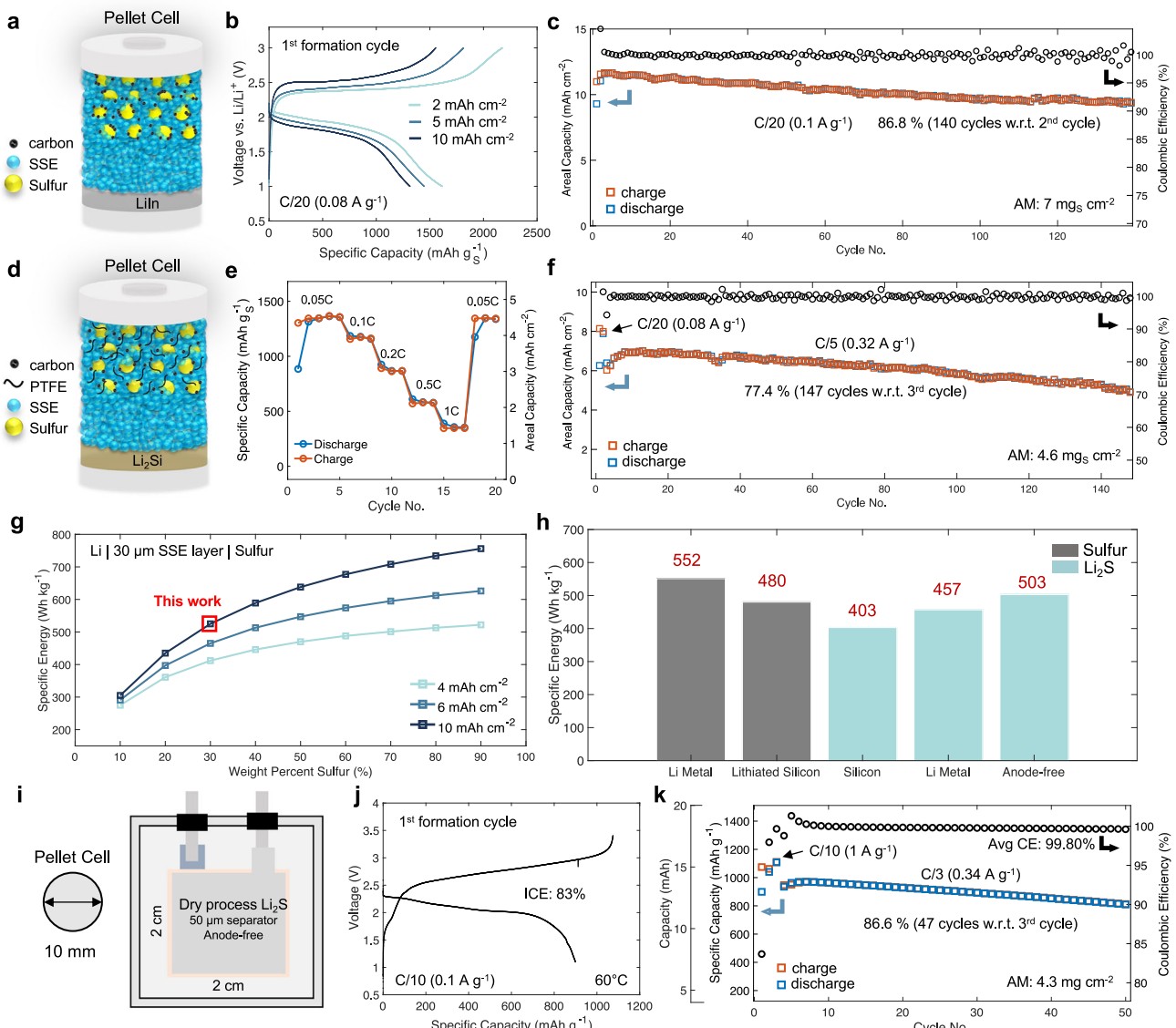

**Fig. 7 | High loading sulfur and Li₂S electrochemical performance with gravimetric energy density outlook. a** Schematic of the sulfur positive electrode half-cell architecture in pellet cells using a Li-In negative electrode. **b** First cycle voltage profiles of cells with increasing areal capacity. Areal capacity was determined by using the average discharge and charge capacity divided by the cell area of 0.785 cm². Sulfur mass loadings of 1.2 mg$_S$ cm⁻², 2.7 mg$_S$ cm⁻², and 7 mg$_S$ cm⁻² were used for 2 mAh cm⁻², 5.5 mAh cm⁻², and 10 mAh cm⁻², respectively. **c** Long term cycling stability at the 11 mAh cm⁻² level evaluated at 25 °C. **d** Schematic of the dry process sulfur positive electrode paired with a lithiated silicon negative electrode in pellet cells. **e** Rate performance at the 5.5 mAh cm⁻² level. **f** Cycling stability at the 7.4 mAh cm⁻² level at C/5 (0.32 A g⁻¹). N/P ratio: 2. Testing temperature for pellet cells: 25 °C ± 1 °C. **g** Theoretical specific energy as a function of sulfur wt% and electrode areal capacity assuming 1600 mAh g⁻¹ discharge capacity, 30 μm SSE layer, and Li metal as the negative electrode. The sulfur positive electrode composition presented in this work is depicted in red. **h** Specific energy comparison between cells using sulfur and Li₂S positive electrodes paired various high-capacity negative electrodes. Values used for these calculations can be found in Table S8. **i** Schematic of the Li₂S/anode-free pouch cell with an areal capacity of 4.7 mAh cm⁻². **j** First cycle voltage profile considering Li₂S as the active mass. **k** Cycling performance evaluated at C/3 (0.34 A g⁻¹) under an isostatic pressure of 10 MPa and testing temperature of 60 °C.

relatively low stack pressures of 10 MPa compared to the pellet-type cells cycled at 75 MPa (Fig. 7k).

The electrochemical performances above showcase the versatility of the positive electrode design methodology, where appropriate synthesis, SSE selection, and optimal electrode microstructure are critical for Li-S chemistry to achieve high utilization and stable cycling. Overall, this approach overcomes the interfacial, kinetic, and chemomechanical challenges associated with Li-S positive electrodes in ASSBs. Our work provides a thorough electrochemical, mechanical, and morphological analysis of the critical features required to enable high loading and practical Li-S batteries.

## Methods

### Materials preparation and composite fabrication

All materials were dried under vacuum at 80 °C if not anhydrous and stored and prepared in an argon-filled glovebox (< 1ppm of H₂O and O₂). The solid-state electrolyte Li₆PS₅Cl (LPSCl, NEI Corp., USA) was used as the separator layer and catholyte. When used as a catholyte, LPSCl was milled in an argon environment at 400 rpm for 2 h with 5 mm spherical yttria stabilized zirconia media using a high energy planetary ball mill (Retsch, Germany). Similar procedures were done for β-Li₃PS₄ (LPS, NEI Corp, USA). For the sulfur positive electrode, elemental sulfur (99.98%, Sigma Aldrich, USA) was either used as

received or milled at 400 rpm for 10 and 24 h for micron and sub-micron particles with a sample to milling media weight ratio of 1:10. For the $Li_2S$ positive electrode, $Li_2S$ was either used as received (99.98%, Sigma Aldrich, USA) or milled following similar procedures as sulfur. For electrochemical evaluation, optimal positive electrode composites were milled for 1 h (unless otherwise specified) at 500 rpm (unless otherwise stated) using a planetary ball mill with a sample to mill media weight ratio of 1:30. Other composite trials were either first milled with carbon and sulfur at 500 rpm for 1 h, followed by hand mixing the SSE or hand mixing all components using a mortar and pestle for 1 h. The sulfur and $Li_2S$ composites consist of 30 wt% active material, 50 wt% LPSCl and 20 wt% of a conductive agent. The LPSCl and LPS composites consist of 80 wt% SSE and 20 wt% conductive agents. The primary conductive agent used in this work was acetylene black (Sigma Aldrich, USA) for cell cycling and CV measurements. Other conductive agents were also used such as vapor grown carbon fiber (Sigma Aldrich, USA) and Ketjen black (EC-600JD, MSE Supplies, USA). For the LCO cells, niobium coated lithium cobalt oxide (LCO, MSE Supplies, USA) was used as received. The LCO positive electrode composite was prepared by hand-mixing with a mortar and pestle and fabricated with a composition of 70 wt% LCO and 30 wt% LPSCl. The Li-In negative electrodes were prepared by vortex mixing stabilized Li metal powders (99.9%, FMC, USA) and indium metal powders (99.99%, Sigma Aldrich, USA) with either a molar ratio of $Li_{0.5}In$ or $Li_1In$ for 5 min. The lithiated silicon negative electrode with a molar ratio of $Li_2Si$ was fabricated in a similar fashion, by vortex mixing μSi (99%, Sigma Aldrich, USA) and stabilized Li metal powders for 3 min, as described by Ham et al.[64]. The silicon negative electrodes were fabricated by casting a 99.9 wt% silicon slurry consisting of μSi particles, 99.5% purity N-methylpyrrolidone, and polyvinylidene fluoride (Sigma Aldrich, USA) on a copper current collector using a doctor blade. The electrode films were dried under vacuum at 80 °C overnight, punched into 10 mm diameter discs and transferred into an argon-filled glove box prior to use, as described by Tan et al.[58]. All materials and electrodes were stored in an argon-filled glovebox at a storage temperature of 22 °C ± 3 °C and used within 1-4 weeks of preparation.

### Dry process positive electrode fabrication

Positive electrode composite powders were mixed with 1 wt% PTFE (Chemours, USA) in a hot mortar and pestle (50–60 °C) until a dough like consistency is formed. The positive electrode composite was then hot rolled (MTI corp., USA) under 60 °C conditions with decreasing thickness until a 300 to 200-micron film was made, correlating to an areal loading of approximately 4.5–6 mAh cm$^{-2}$.

### SSE film and anode-free layer fabrication

For preparing the materials used in the pouch cell, the SSE film consisted of LPSCl (98 wt%) and an acrylate binder (2 wt%) that were mixed in p-xylene (Sigma Aldrich, USA). The resulting mixture was casted on a polyethylene terephthalate film and dried under vacuum at 40 °C overnight. The slurry for anode-free layer was prepared by mixing carbon black (Imerys, USA), silver nanoparticles, and polyvinylidene fluoride (Solvay, Belgium) in N-methylpyrrolidone (Sigma Aldrich, USA) at a weight ratio of 69.75:23.25:7.0 as described by Lee et al.[65]. The slurry was coated onto a 10 μm thick stainless-steel foil using a doctor blade and subsequently dried in a vacuum oven at 100 °C overnight.

### Cell fabrication and electrochemical measurements

Electrochemical measurements were performed in custom 10 mm diameter pellet cells constructed out of Grade 5 titanium plungers and polyether ether ketone (PEEK) dies. For electrochemical measurements in pellet cells, the separator layers were first pressed at 3 tons (375 MPa) for 3 min, resulting in a thickness of 450–500 μm. Positive electrodes and μSi negative electrodes were pressed to 3 tons for 5 min, while the LiIn and $Li_2Si$ negative electrode was pressed to 1 ton

(125 MPa) for 30 seconds. After assembly, the pellet cells were inserted into custom cell holders and hand tightened to 75 MPa. For each electrochemical measurement, three cells were fabricated and tested. EIS and CV measurements were collected using a Biologic SP-300 potentiostat directly after cell assembly. For EIS measurements a voltage amplitude of 30 mV and frequency range of 7 MHz to 100 mHz was used, with 10 points per decade. Ionic conductivity for the solid-state electrolyte and composites was calculated using the following equation $\sigma = \frac{L}{RA}$, where L is the pellet thickness, R is the resistance, and A is the area of the pellet cell (0.785 cm$^2$ for 10 mm cell diameter). The cell configuration used to assess ionic conductivity was Ti|SSE|Ti or Ti|composite|Ti. Equivalent circuit modeling was performed using ZView. For CV measurements, a sweeping rate of 0.1 mV/s was used within the operating voltage range of sulfur (1–3 V vs. Li/Li$^+$). For all electrochemical performance evaluation, pellet cells were cycled under 75 MPa (unless otherwise stated). Galvanostatic cycling measurements were evaluated using Neware Instrument cyclers (CT-4008T). Constant current was applied to cut-off potentials of 1–3 V vs. Li/Li$^+$ for $Li_1In$||Sulfur cells, 0.7–3 V vs. Li/Li$^+$ $Li_2Si$||Sulfur cells, 3.4 V – 1 V vs. Li/Li$^+$ for Si||$Li_2S$ cells, and 3.4 – 1 V vs. Li/Li$^+$ for anode-free||$Li_2S$ pouch cell. All cycling measurements were conducted in an inert glovebox environment at 25 °C ± 1 °C unless otherwise stated. The anode-free||$Li_2S$ pouch cell was cycled in a convection environmental chamber at 60 °C. The designed cell areal capacity was determined by the active mass of the positive electrode multiplied by the corresponding specific capacity (1600 mAh g$^{-1}$ for sulfur and 1100 mAh g$^{-1}$ for $Li_2S$) divided by the cell area (0.785 cm$^2$ for pellet cells and 3.24 cm$^2$ for pouch cell). For cells labeled with their areal capacity, areal capacity was determined by dividing the charge capacity for $Li_2S$ and average discharge capacity for sulfur, by the electrode area. The reported specific currents are the applied currents divided by the active mass of either sulfur or $Li_2S$. In some cases, the experimentally obtained specific capacity is reported by dividing the cell capacity by the total active mass, considering both AM and the SSE within the positive electrode. For the sulfur system, the coulombic efficiency is calculated by dividing the charge capacity by the discharge capacity. For the $Li_2S$ system, coulombic efficiency is reported as the discharge capacity divided by the charge capacity.

### Pouch cell fabrication

Al foil, dry processed $Li_2S$ positive electrode, SSE film and anode-free layer, were stacked, and packed into a pouch. The pouch was vacuum-sealed and pressed to 500 MPa at 80 °C using warm isostatic pressure (WIP).

### Scanning electron microscopy

Scanning electron microscopy (SEM) was performed on a FEI Apreo and/or FEI Scios DualBeam focused ion beam (FIB)/SEM with 5 kV accelerating voltage and 0.1 nA beam current for powders and pellets. Powders, pellets, and cycled electrode samples were prepared in an argon-filled glovebox. Pellets and cycled electrodes were carefully extracted from plunger cells using a polished Ti plunger and mounted to an SEM stub using carbon tape. All SEM samples were transferred using an air-tight transfer arm to avoid any air exposure. For FIB cross-sectional images, milling was done under cryogenic conditions (-180 °C) where Ga was used as an ion beam source. Parameters used for all milling conditions of 30 kV, 65 nA, with the subsequent cross-section cleaning performed with 30 kV, 15 to 7 nA, if necessary.

### X-ray diffraction

XRD measurements were collected over a 5–50° 2θ range on a Bruker ApexII-Ultra CCD microfocus Rotating Anode instrument with Mo K$_\alpha$ (λ = 0.7107 Å) radiation at the UCSD X-Ray Crystallography Department. Samples were prepared in an argon-filled glovebox using 0.7 mm boron capillaries, and flame sealed to ensure air-tight

measurements of sensitive samples. For cycled electrodes, electrodes were extracted using a scalpel and lightly ground to powder using a mortar and pestle for capillary preparation.

## Raman spectroscopy

Raman spectra were collected using a Renishaw inVia upright microscope using 785 nm wavelength laser source. Samples were prepared in an argon-filled glovebox with powders or extracted electrodes placed on 25x25mm glass slides and sealed with Kapton tape for airtightness.

## Thermogravimetric analysis

TGA measurements were conducted using a NETZSCH STA 449 F3 Jupiter Simultaneous Thermal Analyzer. Samples were prepared and loaded in aluminum pans and crimped inside the glovebox for airtight analysis. Aluminum pans were sealed in 20 ml glass vials for transport. During the measurements, nitrogen gas was used to reduce any air exposure. All samples were measured from room temperature to 450 °C using a scan rate of 5 °C/min. Vendor specified resolution is 0.1 μg.

## Transmission electron microscopy

The composite powders were mounted to an airtight cooling holder from Melbuild in an argon-filled glovebox to eliminate any contaminations to the sulfur samples and transferred to the TEM column directly without any air or moisture exposure. The sample was cooled down to cryogenic conditions (~180 °C) and stabilized for additional 30 min before electron beam exposure. (S)TEM results were obtained on ThermoFisher Talos X200 equipped with a Ceta camera operated at 200 kV with low dose capability. The energy dispersive X-ray spectroscopy (EDS) characterization is installed with compositional mapping using 4 in-column SDD Super-X windowless detectors. The data acquisition was operated at low dose condition to minimize any beam damage to the sample.

## X-ray absorption spectroscopy

Tender X-ray absorption spectroscopy (XAS) measurements were conducted at Taiwan Light Source (TLS) beamline 16A1 of the National Synchrotron Radiation Research Center (NSRRC) in Hsinchu, Taiwan. The beamline uses a double-crystal Si (111) monochromator for the photon energy range from 2 to 8 keV. Pristine and cycled electrodes were extracted in pellet form. All samples were sealed in a pouch made of 2.5 um thick Mylar® film inside an argon-filled glovebox to prevent air exposure. Each sample was mounted onto the holder and placed in the measuring chamber at an angle of 45° to the incident X-ray beam. The chamber is constantly purged with He to reduce the X-ray attenuation for at least 45 min before collecting the XAS data. The Sulfur *K*-edge X-ray absorption near-edge structure (XANES) spectra were collected in the total fluorescence yield (TFY) mode using a Lytle detector with a scanning step of 0.2 eV. The photon energy was calibrated to 2472 eV (maximum in the 1st derivative) at the S *K*-edge using elemental sulfur. The XANES spectra background subtraction, normalization, and the Linear Combination Fit (LCF) were performed on Athena software[66].

## Modeling of sulfur electrode geometries

The electrodes structures were stochastically generated using the MATLAB codes from Duquesnoy et al.[67], with the S as spherical particles, and the carbon additives as aggregates. A volume fraction of 10% was dedicated to pores, the S amount ranged from 30 to 60 %, and the volume ratio between the LPSCl and carbon additives was kept constant at 5:2. Three different cases were investigated, Bulk, Micro and Nano, with S radii ranging respectively from 25 to 50 μm, 0.5 to 5 μm, and 0.25 to 0.5 μm. To have a representative volume for each condition, the length of the cubic electrodes was 200 μm for the Bulk, 50 μm for the Micro, and 15 μm for the Nano. For each set of S size and amount, 3 electrodes were generated to obtain statistically relevant observables. The evolution of the active surface area was monitored as the specific surface area, i.e., the ratio between the number of pixels of S in contact with LPSCl and the total number of S pixels. The tortuosity of the LPSCl phase was investigated using TauFactor[68] in MATLAB, and the reported value is the average value of the tortuosity of the electrolyte phase in all directions.

## Finite elements method (FEM) simulations

The electrodes were meshed using the open-access toolbox Iso2Mesh[69] and later imported into COMSOL Multiphysics 6.1. There, using the Solids Mechanics module, the set of parameters and equations in Tables S3 and S4 were set. In the model, the electrode was assumed to be fully compact (no porosity) and the S particles were uniformly lithiated throughout the simulation, leading to a volume expansion made possible with the "Hygroscopic Swelling" node which normally accounts for the volume expansion of solids due to the amount of water. During the simulation, the external boundaries of the electrode were fixed. To determine the analog hygroscopic coefficient of each type of S particles, a 2-D simulation consisting of the exact same model for a single S particle was performed. The hygroscopic coefficient was deemed adequate when the S particle would reach the desired volumetric expansion (controlled here by its radius) at full lithiation. The cases of sub-micro and micron S were investigated through FEM simulations for an AM content of 30 wt% where three electrodes were used for each case, and the value reported in the manuscript are averaged over all three electrodes.

## Data availability

Source data are provided with this paper.

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

## Acknowledgements

This work was supported by the LG Energy Solution – U.C. San Diego Frontier Research Laboratory (FRL) Program (A.C., S.-Y.H., C.H., H.Y., M.V., C.L., D.L., M.-S.S., J.J., J.B.L., and Y.S.M.). A.C. acknowledges the National Science Foundation for having supported their Ph.D. research through the NSF Graduate Research Fellowship Program. The authors (A.C., B.S., and P.R.) would like to acknowledge the UCSD Crystallography Facility. This work was also performed in part at the San Diego Nanotechnology Infrastructure (SDNI) of UCSD, a member of the National Nanotechnology Coordinated Infrastructure, which is supported by the National Science Foundation (Grant ECCS–1542148, A.C., S.-Y.H., S.B., D.C., G.D., and M.V.), along with the use of facilities and instrumentation supported by NSF through the UC San Diego Materials Research Science and Engineering Center (UCSD MRSEC) (Grant DMR-201192). The authors (A.C., C.H., Y.S.M.) would like to acknowledge Prof. Bing Joe Hwang and Ms. Chia-Yu Chang for their help on XAS measurements at the Taiwan Light Source (TLS) beamline 16A1 of the National Synchrotron Radiation Research Center (NSRRC) in Hsinchu, Taiwan. The authors (A.C. and Y.S.M.) thank Dr. Jinkwan Jung for his assistance in sample preparation and cell fabrication.

## Author contributions

A.C. and Y.S.M. conceived the ideas for the study. A.C., J.B.L., and X.W. designed the experiments. B.S. and P.R. collected XRD measurements. S.B. contributed cryo-TEM characterization and investigation. C.H. performed XAS and analysis. A.C., M.V., and D.C. performed SEM-FIB. M.C. performed the electrode modeling and FEM simulations. C.L. and D.L. fabricated and evaluated the pouch cells. J.A.S.O., G.D., S.-Y.H., H.Y., M.V., J.J., M.-S.S., J.B.L., and Y.S.M. participated in the scientific discussion and data analysis. A.C. wrote the manuscript. Y.S.M. supervised the project. All authors discussed the results and commented on the manuscript.

## Competing interests

Y.S.M., A.C., J.B.L., and M.-S.S. declare that two patents were filed for this work through UC San Diego's Office of Innovation and Commercialization and LG Energy Solution, Ltd. The remaining authors declare no competing interests.
