## [Transparent Peer Review file · Nature Communications]

A highly utilized and practical lithium-sulfur positive electrode enabled in all-solid-state batteries

Corresponding Author: Professor Ying Shirley Meng

Version 0:

Reviewer comments:

Reviewer #4

(Remarks to the Author)

The extensive data presented in this manuscript reflect meaningful technical progress in the development of all-solid-state lithium-sulfur batteries (ASSLSBs). However, given the number and complexity of the experiments, several aspects of the experimental design lack sufficient detail, and further clarification is necessary. Specific concerns are outlined below.

1. Influence of mechanical milling on SSE properties and S-SSE Interface. Mechanical milling appears to play a critical role in altering the physicochemical properties of the SSE and the formation of the S-SSE interface. Milling intensity determines the amount of energy imparted to the material, and while hand mixing can be regarded as a low-energy baseline, directly comparing it with high-energy milling at 500 rpm offers only a limited perspective. Please include a discussion, supported by appropriate electrochemical tests and literature references, on how varying milling intensities affect interface formation and electrochemical performance.
2. Electrochemical redox mechanism during cycling of 1-step milled ASSLSBs. What are the dominant redox reactions occurring at each potential during cycling in the 1-step milled ASSLSB? Furthermore, does $\text{Li}_3\text{PS}_{4+n}$ contribute to the overall capacity? Since the reversible redox reaction of the SSE appears to occur from the second cycle (Fig. 3a), the voltage profile and dQ/dV analysis beyond the second cycle should be done to clarify the electrochemical behavior of the proposed ASSLSBs. It seems that the SSE's redox activity occurs at higher potentials than that of sulfur. (Fig. 3a and 3b)
3. Stability of $\text{Li}_3\text{PS}_{4+n}$ during cycling. Beyond confirming the formation and functionality of $\text{Li}_3\text{PS}_{4+n}$ in the initial stage, how does this material evolve during cycling? Is it retained or transformed in later cycles? Please clarify and provide evidence for these changes.
4. Stability and structural evolution of milled sulfur. It is stated that micron-sized and sub-micron sulfur were obtained via 400 rpm milling, yet the final cathode composite is fabricated through 500 rpm milling. Is it reasonable to assume that the particle size from the pre-milling step is retained after subsequent composite fabrication? Also, does milling affect only the particle size, or are there other changes such as in crystallinity or bonding structure of sulfur? These aspects are critical to cathode design, and a more thorough explanation is required to convincingly support the role of milled sulfur.
5. Clarification of cell compositions used in electrochemical tests. Due to the large number of cathode composite configurations and corresponding electrochemical data, it is difficult to follow the overall flow. Please clearly summarize and specify the cell configurations used in each data panel to improve readability and interpretation.

Reviewer #5

(Remarks to the Author)

In this work, the authors report the fabrication of high-performance all-solid-state sulfur/Li₂S cathodes using a one-step high-energy ball milling method. This process induces the formation of sulfur-rich thiophosphate phases ($\text{Li}_3\text{PS}_{4+n}$), which

facilitate enhanced interfacial ion transport. The study presents a comprehensive investigation into the structural evolution of cathode materials during cycling, the redox activity of cathode components, the influence of electrode microstructure (particularly sulfur particle size) on battery performance, volume-pressure evolution during electrochemical cycling, and the overall electrochemical performance of the all-solid-state batteries. The performance demonstrated in both half-cells and practical full pouch cells is promising and offers insights into the development of practical solid-state lithium-sulfur batteries. However, several critical issues need to be addressed before publication:

1. The areal sulfur loading should be clearly specified for all electrochemical data, particularly in Fig. 2a, Fig. 3, and Fig. 5e.
2. For improved clarity, the authors should provide the calculated ionic conductivity values of the cathode composites and Li₆PS₅Cl (LPSCI) prepared under different milling conditions, in addition to the EIS spectra shown in Fig. S3c,f.
3. As X-ray diffraction (XRD) patterns only indicate the crystal structure of the materials, they are unable to identify the amorphous sulfur-rich interlayer (Li₃PS_{4+n}) after cycling (line 331). Instead, Raman spectroscopy should be conducted to validate the existence of the interlayer phase after cycling.
4. Although sulfur with different particle sizes was used, the high-energy ball milling process shall further reduce their particle sizes. Therefore, SEM-EDS mapping of the pristine cathode surface is recommended to examine the actual sulfur particle size after fabrication.
5. It is unclear in Fig. 6c and d which curves correspond to pressure changes, and which represent voltage during cycling. The figure should be redrawn to improve clarity. Additionally, due to the relatively small pressure changes in all-solid-state lithium-sulfur batteries, the authors are encouraged to reduce the y-axis range for pressure to better illustrate the pressure evolution trends during cycling.
6. It is recommended to use the same color (yellow) for sulfur particles in fig. 7d as in Fig. 7a to maintain consistency.
7. The current densities are reported in C-rate, but it is unclear how 1C is defined. Given the varying cathode compositions (e.g., sulfur, Li₂S, and LPSCI), all current densities are encouraged to be reported in A g⁻¹ based on the total weight of sulfur/Li₂S and LPSCI.
8. The authors claim the formation of both Li₃PS₄ and Li₄P₂S₆ based on a single XRD crystalline peak (Fig. 4b). This is not sufficient evidence. Additional techniques such as solid-state NMR or Raman spectroscopy should be employed to verify the presence of Li₄P₂S₆. Moreover, prior literature reports Li₄P₂S₇ rather than Li₄P₂S₆ as an oxidation product of LPSCI (Nature Materials 2020, 19, 428–435.) and this discrepancy should be addressed.
9. The error bars in the figures should be clearly defined. The number of replicates used to generate the error bars should be specified in the respective figure captions.
10. The figure caption for Fig. 5f should explicitly state whether the EIS spectra were collected at the charged or discharged state.
11. On line 354, what is L_{4.1}PS₄ phase? Is it Li₄PS₄ or Li_{4.1}PS₄ phase?

Regarding the questions raised by previous Reviewer #2, the authors have addressed some of them but the following still needs further elucidation:

1. Comment 3 and 8 by Reviewer #2

The authors claim the formation of Li₄P₂S₆, but as previously noted, this has not been conclusively demonstrated. Additional characterization is needed to confirm its presence.

2. Comment 6 by Reviewer #2 - Redox Reactions

In the response, the authors state that the cathode contains 43.5 wt% LPSCI, 6.5 wt% Li₃PS_{4+n}, and 23.5 wt% sulfur, and that the capacity contribution from Li₃PS_{4+n} is marginal.

However, this analysis is flawed. First, if 6.5 wt% of sulfur has reacted, the corresponding Li₃PS_{4+n} must be more than 6.5 wt% due to the added mass of the thiophosphate framework. Second, the chain length of sulfur in Li₃PS_{4+n} is unknown, making accurate quantification via TGA alone infeasible. Third, the theoretical capacity of Li₃PS_{4+n} (depending on the sulfur chain length) can exceed 500 mAh g⁻¹, making its capacity contribution significant (see Angew. Chem. Int. Ed., 2013, 52, 7608–7611).

3. Comment 9 by Reviewer #2 - Discrepancy Between XANES and XRD Results

The quantification of the LPSCI using XANES is intriguing. However, as stated by reviewer #2 the XANES data in Fig. 4e suggest complete decomposition of LPSCI after discharge, whereas XRD (Fig. 4b) still shows strong LPSCI peaks. The authors attribute this to differences in sulfur loading and the shallow probe depth of XAS but provide insufficient supporting evidence. Given the high LPSCI content, full reduction to LPS is unlikely despite the low areal sulfur loading. To support this claim, the authors should include the corresponding electrochemical data (e.g., voltage profiles, dQ/dV curves) for the cells used in the XRD and XAS experiments in the Supplementary Information. Testing conditions and cell compositions should be clearly specified, and capacity contributions quantified. Additionally, the manuscript should include a concise explanation, with a more detailed discussion in the Supplementary Information, regarding the accuracy and limitations of the XANES quantification.

4. Comment 11 by Reviewer #2 - XPS

XPS should be removed from the Author Contribution section.

Reviewer #6

(Remarks to the Author)

I think the authors have adequately addressed the concerns from the previous reviewers for publication in this journal.

Version 1:

Reviewer comments:

Reviewer #4

(Remarks to the Author)

All of my comments have now been addressed.

Reviewer #5

(Remarks to the Author)

I would like to thank the authors for their extensive efforts in addressing all the previous questions. I am in favor of its publication. In addition, please correct the typo, "LiCO₂", in Fig. 6c and d to "LiCoO₂."

Reviewer Comments

Note: Reviewer's comments are shown in **black**, and our responses to the comments are shown in **blue**.

Reviewer #4

The extensive data presented in this manuscript reflect meaningful technical progress in the development of all-solid-state lithium-sulfur batteries (ASSLSBs). However, given the number and complexity of the experiments, several aspects of the experimental design lack sufficient detail, and further clarification is necessary. Specific concerns are outlined below.

Comment 1: Influence of mechanical milling on SSE properties and S-SSE Interface. Mechanical milling appears to play a critical role in altering the physicochemical properties of the SSE and the formation of the S-SSE interface. Milling intensity determines the amount of energy imparted to the material, and while hand mixing can be regarded as a low-energy baseline, directly comparing it with high-energy milling at 500 rpm offers only a limited perspective. Please include a discussion, supported by appropriate electrochemical tests and literature references, on how varying milling intensities affect interface formation and electrochemical performance.

Response: We thank the reviewer for the constructive feedback and opportunity to improve our synthesis investigation. We have increased the synthesis scope to include different milling intensities to demonstrate its effect on the S-SSE interface and electrochemical performance. With 500 rpm being the maximum for our planetary ball mill, this limits our upper milling intensity parameter. Additionally, with 1 hour being our ideal milling duration, it is not advantageous to increase milling times as this requires more energy and has already been demonstrated to cause LPSCI to degrade. Therefore, we have extended the investigation to include milling intensities of 300 rpm and 400 rpm. This is done to investigate the milling intensity dependence on S-SSE interface formation and sulfur utilization. Cells were fabricated in an identical fashion to the 500rpm case, using a Li-In anode. Their electrochemical performance, compared to our ideal synthesis protocol, is shown below.

The 1st formation cycle is normalized by the sulfur mass, highlighting the clear discrepancies in utilization when using the different milling intensities. The composites fabricated at 300 and 400 rpm deliver a specific capacity of 565 mAh g⁻¹ and 900 mAh g⁻¹, respectively. Reducing the milling intensity was found to directly impact sulfur utilization, suggesting that the lower milling intensity is likely insufficient to fully induce the reaction between sulfur and LPSCl, leading to a lower wt.% of reacted sulfur. The reacted sulfur in the Li₃PS_{4+n} phase was found to be ionically conductive. A higher percentage of unreacted bulk sulfur limits the ionic conductivity of the composite, evidenced by the increased cell polarization.

In all composites, the conversion efficiency was over 100%, indicating that LPSCl redox activity has been activated. The 2nd cycle is also shown below. Voltage profiles were normalized by the active mass (sulfur and LPSCl) to evaluate the impact milling intensity has on the capacity contribution of LPSCl. All composites also deliver a higher 2nd discharge capacity than the initial, indicating that LPSCl redox is reversible despite its lower capacity contribution for the 300 and 400 rpm cases.

In summary, high milling intensities are required to facilitate the interfacial reaction between sulfur and LPSCl. Lower rpms can be employed to activate LPSCl redox activity and enable reversible cycling as a consequence of lower sulfur utilization due to the incomplete formation of the S-SSE interface.

Additional discussion has been added to the manuscript with two new references.

“Cathode composite architecture including conductive interfaces are one of the critical components to enable high performing Li-S ASSBs. The reactivity between the active materials

and the sulfide SSE was investigated using a one-step mechanochemical milling procedure, where high milling intensities were used for a short duration. The milling intensity is a complex parameter that influences the reaction kinetics by tuning the collision frequency between particles. High intensities have been found to accelerate reaction products often leading to shorter process times.^{34,35} These two parameters were found to be critical in creating conductive interfaces between sulfur and the sulfide SSE without degrading ion conduction pathways within the composite.”

34. C. Burmeister, L. Titscher, S. Breitung-Faes, A. Kwade, Dry grinding in planetary ball mills: Evaluation of a stressing model. *Advanced Powder Technol.* 29, 191-201 (2017).

35. M. Hofer, M. Grube, C.F. Burmeister, P. Michalowski, S. Zellmer, A. Kwade, Effective mechanochemical synthesis of sulfide solid electrolyte Li_3PS_4 in a high energy ball mill by process investigation. *Advanced Powder Technol.* 34, 104004 (2023).

The following discussion was added to line 203 to accompany the new Supplementary Fig. S3.

“The bonding between the thiophosphate units of LPSCl and sulfur, facilitated by the high energy synthesis used in this work, ensures intimate contact of the sulfur/SSE interface with high sulfur utilization shown in Fig. 2a. A high milling intensity is required to induce the interfacial reaction between sulfur and LPSCl. Sulfur utilization was found to decrease when using lower milling intensities (fig. S3), suggesting incomplete formation of the $\text{Li}_3\text{PS}_{4+n}$ phase.”

New Fig. S3

Fig S3. Evaluation of lower milling intensities using the one-step milling process in Li-In half cells. a) 1st formation cycle normalized by sulfur mass. 1st and 2nd voltage profiles normalized by total active mass for b) 300 rpm, c) 400 rpm, and d) 500 rpm.

Additional emphases to the role of milling intensity were added to line 221 of the manuscript after the ionic conductivity measurements, where long milling durations were explored.

“These results highlight the importance of limiting milling durations, where high milling intensities used for short durations were found to be sufficient to complete the interfacial reaction between sulfur and LPSCl while preserving the ionic conductivity of the SSE and cathode composite.”

Comment 2: Electrochemical redox mechanism during cycling of 1-step milled ASSLSBs. What are the dominant redox reactions occurring at each potential during cycling in the 1-step milled ASSLSB? Furthermore, does $\text{Li}_3\text{PS}_{4+n}$ contribute to the overall capacity? Since the reversible redox reaction of the SSE appears to occur from the second cycle (Fig. 3a), the voltage profile and dQ/dV analysis beyond the second cycle should be done to clarify the electrochemical behavior of the proposed ASSLSBs. It seems that the SSE's redox activity occurs at higher potentials than that of sulfur. (Fig. 3a and 3b).

Response: The reviewer has brought up some insightful points and great questions. We appreciate the opportunity to clarify and will address each question point by point.

1. What are the dominant redox reactions occurring at each potential during cycling in the 1-step milled ASSLSB?

From our work, we show that the 1-step high energy milling process facilitates sulfur bonding between sulfur and sulfur in LPSCl, where 6.5 wt.% of the composite (~ 20 % of sulfur) undergoes an interfacial reaction with LPSCl. The remaining 23.5 wt.% is unreacted sulfur. To better understand the dominant redox reactions, cells with decreasing sulfur wt.% were fabricated. Capacity was normalized by the total active mass (sulfur and LPSCl) to highlight individual capacity contributions.

During the initial discharge (**Region I**), from OCV to 1.3V, the dominant reaction is the reduction of sulfur from the unreacted bulk sulfur in the composite. For cells with higher sulfur wt.%, most of the discharge capacity is obtained at this potential, meaning this is from unreacted bulk sulfur. This reduction process starts at 2.1V, commonly observed for Li-S solid-state cells attributed to the solid-solid reduction to Li_2S . **Region II** starts at 1.3V where the reacted sulfur in $\text{Li}_3\text{PS}_{4+n}$ phase begins to reduce. This reduction peak is easily identifiable in the dQ/dV curves for all cells, but prominent for the cell with lower sulfur to LPSCl wt.% due to its higher capacity contribution compared to unreacted sulfur.

The early reduction of the sulfur rich LPS phase is expected and follows a similar redox behavior of LPS, which reduces at a higher potential than LPSCl. The Cl in LPSCl increases its reduction stability¹, which is why the LPSCl/C cell reduces at a lower potential compared to LPS. This is also supported by the behavior of the SSE/C cells shown below, where LPS exhibits a narrower electrochemical stability window. The dQ/dV curves of LPSCl/C (no sulfur) and various S/LPSCl wt.% highlight the S and LPSCl interaction, which creates a less stable LPS phase that readily reduces before LPSCl in the 1st discharge. Since we observe that LPS reduces at a higher potential and delivers a higher discharge capacity, we would expect the reduction of this phase to dominate at this potential until near 1.13V, where LPSCl begins to reduce.

New Fig. S12

We hypothesize that the reacted sulfur in sulfur rich $\text{Li}_3\text{PS}_{4+n}$ phase actively participates in reversible redox behavior following similar behavior to LPS. The additional sulfur in PS_4^{3-} increases its functional capacity contribution and reduces its stability. Both the sulfur rich LPS phase and LPSCl are expected to reversibly form LPS.

Region III is from 2V till 2.7V where Li_2S oxidation is dominated by the conversion of the formed Li_2S from the unreacted bulk sulfur. However, both LPSCI and LPS phases will contribute capacity at this potential as shown in the dQ/dV curves. Higher wt.% of sulfur will contribute higher capacity at this potential (meaning there was more of the Li_2S phase due to the higher wt.% of sulfur within the composite that reduced). **Region IV** starts at 2.7V where oxidation capacity is obtained from both the $\text{Li}_3\text{PS}_{4+n}$ and LPSCI phases, forming elemental sulfur and LPS. The additional oxidation capacity is recoverable upon the subsequent discharge.

LPSCI forms a reversible LPS phase after the first cycle. $\text{Li}_3\text{PS}_{4+n}$ actively participates in the redox reactions. We are unable to quantify the sulfur chain length in the $\text{Li}_3\text{PS}_{4+n}$ phase, but $\text{Li}_3\text{PS}_{4+3}$ is estimated as the starting phase based on the calculations performed below.

I. Y. Liu, et al., Revealing the Impact of Chlorine Substitution on the Crystallization Behavior and Interfacial Stability of Superionic Lithium Argyrodites, Adv. Funct. Mater. 2207978 (2022).

We have clarified and elaborated the discussion on the dominant redox reactions within the manuscript.

“Four main regions are labeled in the dQ/dV curves. Region I highlights the dominant reaction during the discharge, attributed to the reduction of sulfur. Cells with higher sulfur wt.% deliver most of their discharge capacity at this potential, meaning this is from unreacted bulk sulfur. Region II highlights the beginning of SSE reduction. Starting at 1.3V, the reacted sulfur in $\text{Li}_3\text{PS}_{4+n}$ begins to reduce. This reduction peak is identifiable in the dQ/dV curves for all cells, but mostly prominent for the cell with the lowest sulfur wt.% due to a higher capacity contribution from this phase. The $\text{Li}_3\text{PS}_{4+n}$ phase is expected to follow similar redox as LPS, which reduces at a higher potential than LPSCI (fig. S12). Therefore, we would expect the reduction of this phase to dominate until 1.15V, where LPSCI begins to reduce. We can estimate their individual discharge capacity contributions based on the reduction potentials obtained from dQ/dV . From OCV to 1.3V, 1 mAh is obtained, as estimated above. The remaining capacity from 1.3 to 1.15V can be attributed to $\text{Li}_3\text{PS}_{4+n}$, which is responsible for 7.5% of the overall discharge capacity. The remaining

capacity obtained from 1.15 to the lower voltage cutoff of 1V is from LPSCI, which contributes 9.2%. The specific capacity of the reacted sulfur is estimated to be 553 mAh g⁻¹ during the discharge, likely existing in the Li₃PS_{4+n} phase after synthesis. These calculations are presented in table S2. The additional sulfur in the PS₄³⁻ framework increases its functional capacity and reduces the electrochemical stability of the SSE. Region III is attributed to Li₂S oxidation. Higher wt.% of sulfur will contribute higher capacity at this potential since more Li₂S was formed during the discharge and is now being oxidized. The cell with just LPSCI and carbon will contribute capacity at this potential, attributed to LPS oxidation as investigated from previous work by Tan et al.²⁶. Region IV starts at 2.7V where SSE oxidation occurs. This additional oxidation capacity is recoverable upon the subsequent discharge, supporting the formation of a reversible redox-active phase.”

26. D. H. S. Tan, E. A. Wu, H. Nguyen, Z. Chen, M. A. T. Marple, J. M. Doux, X. Wang, H. Yang, A. Banerjee, Y. S. Meng, Elucidating Reversible Electrochemical Redox of Li₆PS₅Cl Solid Electrolyte. *ACS Energy Lett.*, 2418–2427 (2019).

2. Furthermore, does Li₃PS_{4+n} contribute to the overall capacity?

Yes, the sulfur rich conductive phase does contribute capacity. We calculated the estimated capacity based on known masses of each component within the cathode composite and compared it to results obtained experimentally.

As a starting point, we used a model cell composition. Our initial electrochemical results exhibited an areal capacity of 1.6 mAh cm⁻² which used a total cathode composite mass of 2.5 mg. The cathode consists of 23.5 wt.% unreacted sulfur, 6.5% reacted sulfur, 50 wt.% LPSCI, and 20 wt.% carbon. A typical cell with this loading delivers 1.2 mAh and 1.60 mAh after the discharge and charge. The table below shows the expected capacity if every component fully contributes. The specific capacity used for unreacted and reacted sulfur is 1675 mAh g⁻¹. The specific capacity used for LPSCI was obtained experimentally using cells with a LPSCI/C cathode, which was 115 mAh g⁻¹ and 355 mAh g⁻¹ during the discharge and charge. For the S/LPSCI/C cell, 1.38 mAh and 1.70 mAh is estimated after the discharge and charge. These capacities are close to what was obtained experimentally.

Expected Capacity Based on Mass for each Component				
	Unreacted Sulfur (23.5 wt.%) (mAh)	Reacted Sulfur in Li ₃ PS _{4+n} (6.5 wt.%) (mAh)	LPSCI (50 wt.%) (mAh)	Total Capacity (mAh)
Discharge	1	0.25	0.13	1.38
Charge	1	0.25	0.44	1.70

Assuming full utilization of the unreacted sulfur mass, 1 mAh is estimated. Since the total discharge capacity is 1.2 mAh, the remaining 0.2 mAh is either from the reacted sulfur in $\text{Li}_3\text{PS}_{4+n}$, LPSCI, or a mixture of both.

Capacity Estimated based on Electrochemistry				
	Unreacted Sulfur (23.5 wt.%) (mAh)	Reacted Sulfur in $\text{Li}_3\text{PS}_{4+n}$ (6.5 wt.%) (mAh)	LPSCI (50 wt.%) (mAh)	Total Capacity (mAh)
Discharge	1	0.20		1.20
Charge	1	0.16	0.44	1.60

After the charge, a total cell capacity of 1.6 mAh is obtained. Assuming similar electrochemical behavior as for the LPSCI/C cells (80/20 wt.%), LPSCI is expected to deliver a charge capacity of 355 mAh g^{-1} , which equates to 0.44 mAh using a known mass of 1.25 mg of LPSCI within the composite. To estimate the remaining capacity for $\text{Li}_3\text{PS}_{4+n}$, the unreacted sulfur and LPSCI is subtracted from the total cell capacity, equating to 0.16 mAh. Using the dQ/dV curves during the initial discharge, we can estimate the discharge capacity based on the potentials that $\text{Li}_3\text{PS}_{4+n}$ and LPSCI reduces. $\text{Li}_3\text{PS}_{4+n}$ phase reduces at 1.3V while LPSCI reduces at 1.15V. Using these reduction potentials from the dQ/dV curves, we can estimate the capacity obtained. From OCV to 1.3V, 1 mAh is obtained as estimated above. The remaining capacity from 1.3 to 1.15 results in the following estimation.

Capacity Estimated based on potentials from dQ/dV				
	Unreacted Sulfur (23.5 wt.%) (mAh)	Reacted Sulfur in $\text{Li}_3\text{PS}_{4+n}$ (6.5 wt.%) (mAh)	LPSCI (50 wt.%) (mAh)	Total Capacity (mAh)
Discharge	1	0.09	0.11	1.20
Charge	1	0.16	0.44	1.60

The specific capacity of sulfur in $\text{Li}_3\text{PS}_{4+n}$ is estimated to be 553 mAh g^{-1} during the discharge and 984 mAh g^{-1} during the charge. Since we validated the existence of this phase after cycling, we can conclude that its redox is reversible although a higher capacity is expected during the charging process. Using these estimations, we can assign capacity contributions for each component.

Capacity Contribution (%)				
	Unreacted Sulfur (23.5 wt.%) (mAh)	Reacted Sulfur in $\text{Li}_3\text{PS}_{4+n}$ (6.5 wt.%) (mAh)	LPSCI (50 wt.%) (mAh)	Total Capacity (mAh)
Discharge	83.3	7.5	9.2	1.20
Charge	62.1	10	27.5	1.60

Table S2. Estimated capacity contributions between unreacted sulfur, reacted sulfur in $\text{Li}_3\text{PS}_{4+n}$, and LPSCI.

Expected Capacity Based on Mass for each Component				
	Unreacted Sulfur (mAh)	Reacted Sulfur in $\text{Li}_3\text{PS}_{4+n}$ (mAh)	LPSCI (mAh)	Total Capacity (mAh)
Discharge	1	0.25	0.13	1.38
Charge	1	0.25	0.44	1.70
Capacity Estimated based on Electrochemistry				
Discharge	1	0.20		1.20
Charge	1	0.16	0.44	1.60
Capacity Estimated based on potentials from dQ/dV				
Discharge	1	0.09	0.11	1.20
Charge	1	0.16	0.44	1.60
Capacity Contributions (%)				
Discharge	83.3	7.5	9.2	1.20
Charge	62.1	10	27.5	1.60

These estimations have been added to the supplementary information as Table S2, with additional discussion regarding the capacity contribution of the reacted sulfur in the $\text{Li}_3\text{PS}_{4+n}$ phase.

“In this system, three active materials exist: bulk LPSCI, the formed interlayer phase ($\text{Li}_3\text{PS}_{4+n}$) consisting of sulfur that has reacted with LPSCI, and unreacted elemental sulfur. TGA was used to quantify the reacted and unreacted sulfur mass, where 23.5 out of the 30 wt.% of sulfur remains unaltered after synthesis. Using the known sulfur and LPSCI mass, we can estimate the capacity contributions of the active materials and compare to what was obtained experimentally. For a 1.6 mAh cm^{-2} cell using a total composite mass of 2.5 mg, 23.5 wt.% unreacted sulfur is expected to deliver 1 mAh (using 1675 mAh g^{-1}). Therefore, during the discharge, unreacted bulk sulfur is estimated to contribute 83% of the capacity with the remainder attributed to LPSCI and $\text{Li}_3\text{PS}_{4+n}$ reduction. To isolate the capacity expected from just LPSCI, cells with just LPSCI and carbon were fabricated using a Li-In alloy as a lithium source.”

“Subtracting the expected LPSCI and unreacted bulk sulfur capacity from the total cell capacity obtained after the charge leaves the remaining capacity to be assigned to reacted sulfur in $\text{Li}_3\text{PS}_{4+n}$. This equates to a capacity contribution of 10% for the reacted sulfur and 27.5% for LPSCI after the charge, suggesting that the reacted sulfur in $\text{Li}_3\text{PS}_{4+n}$ and LPSCI is redox-active and actively participates capacity during cycling.”

“We can estimate their individual discharge capacity contributions based on the reduction potentials obtained from dQ/dV . From OCV to 1.3V, 1 mAh is obtained, as estimated above. The remaining capacity from 1.3 to 1.15V can be attributed to $\text{Li}_3\text{PS}_{4+n}$, which is responsible for 7.5% of the overall discharge capacity. The remaining capacity obtained from 1.15 to 1V is from LPSCI, which contributes 9.2%. The specific capacity of the reacted sulfur is estimated to be 553 mAh g^{-1} during the discharge, likely existing in the $\text{Li}_3\text{PS}_{4+3}$ phase after synthesis. These calculations are presented in table S2. The additional sulfur in the PS_4^{3-} framework increases the functional capacity and reduces the electrochemical stability of the SSE.”

- Since the reversible redox reaction of the SSE appears to occur from the second cycle (Fig. 3a), the voltage profile and dQ/dV analysis beyond the second cycle should be done to clarify the electrochemical behavior of the proposed ASSLSBs. It seems that the SSE's redox activity occurs at higher potentials than that of sulfur. (Fig. 3a and 3b).

Response: The reviewer has made an insightful observation regarding the SSE redox activity occurring at higher potentials than that of sulfur. We have addressed this discrepancy above, where LPSCI will reduce at lower potentials than that of sulfur and the reacted sulfur in $\text{Li}_3\text{PS}_{4+n}$. This is because sulfur reduces at 2.1V, with most of its capacity obtained at the 1.9V plateau. The $\text{Li}_3\text{PS}_{4+n}$ phase is expected to follow similar redox behavior as LPS, which reduces at higher potentials than that of LPSCI. This is illustrated in the dQ/dV curves where cells with 10 wt.% of sulfur clearly highlight this reduction peak due to the larger capacity contribution of the $\text{Li}_3\text{PS}_{4+n}$ phase. This reduction peak is still evident for cells with 20 and 30 wt.% sulfur, albeit with smaller capacity contribution.

As suggested by the reviewer, we also performed dQ/dV analysis on the 3rd cycle for all cells. This better highlights the reversible LPS phase formed by both LPSCl and the sulfur rich interlayer. The LPSCl/C cell exhibits reversible redox behavior centered near the reduction and oxidation potential of sulfur, which has previously made deconvoluting this behavior challenging. In the 3rd cycle, clear evidence of SSE redox and oxidation is present in all cells at 1.3 V and 2.7V.

We have updated the main Figure 3 by adding the dQ/dV curves of the 3rd cycle, which better highlights the reversible redox contribution from LPSCl and Li₃PS_{4+n} phase.

Fig. 3 | LPSCI redox activity and capacity contribution. Voltage profiles of LPSCI/C Li-In half cells evaluated under (a) sulfur voltage limits. b, Cyclic voltammetry of sulfur and LPSCI cathode composites with a Li-In anode. c, dQ/dV curves of the 1st and (d) 3rd cycle for cells with decreasing sulfur wt.% (S/LPSCI/C). The areal loading of sulfur was 0.3 mg cm^{-2} , 0.84 mg cm^{-2} , and 1 mg cm^{-2} corresponding to a total active mass of 2.8 mg cm^{-2} , 2.9 mg cm^{-2} , and 2.5 mg cm^{-2} for 10 wt.%, 20 wt.%, and 30 wt.% sulfur composites, respectively.

The following discussion has been added to manuscript.

“This additional oxidation capacity is recoverable upon the subsequent discharge, supporting the formation of a reversible redox-active phase. This is highlighted in the dQ/dV curves for the 3rd cycle (Fig. 3d), where reversible SSE redox occurs at similar potentials as sulfur conversion. Despite a portion of the sulfur reacting with LPSCI, sulfur utilization within each of these cells can be estimated. Cathodes with 30 wt.% sulfur are 84% utilized during the 1st discharge. Sulfur utilization significantly increases for lower sulfur weight percentages, increasing to above 90% for the cathode with 10 wt.% sulfur. These estimations are listed in table S3.”

Comment 3: Stability of $\text{Li}_3\text{PS}_{4+n}$ during cycling. Beyond confirming the formation and functionality of $\text{Li}_3\text{PS}_{4+n}$ in the initial stage, how does this material evolve during cycling? Is it retained or transformed in later cycles? Please clarify and provide evidence for these changes.

Response: The reviewer has asked a great question. The sulfur rich $\text{Li}_3\text{PS}_{4+n}$ phase was originally identified by observing a redshift of the E_2 symmetric S-S bending at 152 cm^{-1} to shorter wavelengths. The degree of the shift is influenced by the sulfur chain length in $\text{Li}_3\text{PS}_{4+n}$, with shorter sulfur chains corresponding to lower wavenumbers, as reported by Lin, et al., *Angewandte Chemie - International Edition*, **52**, 7460–7463 (2013). To gain more insight on how the $\text{Li}_3\text{PS}_{4+n}$ phase can evolve or be preserved with cycling, Raman spectroscopy was conducted on a cycled sulfur cathode after 5 cycles and 100 cycles.

The $\text{Li}_3\text{PS}_{4+n}$ phase was confirmed to be retained after both short- and long-term cycling. A clear redshift is observed in the Raman spectra for all cathode samples, with the symmetric S-S bending region being identified at 150 cm^{-1} . This is comparable to the peak position for the cathode composite directly after synthesis. The retainment of this conductive phase helps explain the long cycle life and sustained sulfur utilization obtained with this cathode system.

The following figure has been added to the Supplementary Information as Fig. S14.

New Fig. S14

Fig. S14. Raman spectra focusing on the symmetric S-S bending region of sulfur cathodes after cycling.

The following discussion has been added to the manuscript. *“The high charge capacity yet low peak intensity and amorphous background indicate that the reformed sulfur is amorphous as in the pristine state and suggests the sulfur rich interlayer is preserved with cycling. This was confirmed by conducting Raman spectroscopy on the cycled sulfur cathode, where a redshift to lower wavenumbers at the S-S bending region is observed (fig. S14), as previously shown for the pristine composite.”*

Comment 4: Stability and structural evolution of milled sulfur. It is stated that micron-sized and sub-micron sulfur were obtained via 400 rpm milling, yet the final cathode composite is fabricated through 500 rpm milling. Is it reasonable to assume that the particle size from the pre-milling step is retained after subsequent composite fabrication? Also, does milling affect only the particle size, or are there other changes such as in crystallinity or bonding structure of sulfur?

These aspects are critical to cathode design, and a more thorough explanation is required to convincingly support the role of milled sulfur.

Response: The reviewer brings up a good point regarding the role of pre-milled sulfur. It should be clarified that two different milling parameters beyond milling intensity were used, which is the sample to milling media weight ratio. A higher milling media to sample weight ratio was used for composite synthesis (30:1), which induces more energy into the system due to a higher collision frequency between the milling media and sample. A lower milling media to sample weight ratio (10:1) was used to reduce the sulfur particle size prior to composite synthesis. This lower weight ratio induces more frictional stress from particle-to-particle interaction, leading to more breakage. This distinction has been added within the methods section and shown below.

“For the sulfur cathode, elemental sulfur (99.98%, Sigma Aldrich) was either used as received or milled at 400 rpm for 10 and 24 hours for micron and sub-micron particles with a sample to milling media weight ratio of 1:10.”

“For electrochemical evaluation, optimal cathode composites were milled for 1 hour (unless otherwise specified) at 500 rpm (unless otherwise stated) using a planetary ball mill with a sample to milling media weight ratio of 1:30.”

Pre-milling sulfur was found to induce more interfacial contact between the SSE and carbon, leading to higher utilization and better rate performance. We were also curious about the impact of high energy milling during synthesis had on particle morphology and size. SEM was first conducted on the bulk sulfur powders and composite after synthesis. We broke up the previous Fig. S16 to help with the flow for the addition of SEM-EDS mapping shown below. Fig. S22 shows that the Bulk sulfur exhibits a wide particle size distribution from 100 to 10 microns. After composite synthesis, a large sulfur particle is identified and surrounded by smaller LPSCI particles (fig. S22). This distinction was made due to their differences in contrast (differences in atomic number between sulfur and LPSCI) from the backscattered electron detector. This large sulfur particle is near 50 μm in diameter, exhibiting a comparable size prior to composite synthesis (fig. S22a).

Fig. S22. Scanning electron microscopy (SEM) images of unmodified sulfur and composite. Powders of a) Bulk Sulfur as received and b) composite after synthesis. c) Top view after cathode composite fabrication.

SEM-EDS mapping was also conducted on another batch of composite. We did find EDS mapping to be challenging on the composite since LPSCI will contribute to the sulfur signal. The

Cl and P K mapping was also included but was still hard to discern the particles due to the good composite distribution and interfacial bonding between sulfur and the SSE. However, the P and Cl mapping show lower intensity at the location of the highlighted sulfur particle. These results are shown below.

A large sulfur particle was identified and labeled in the above image, exhibiting a much darker contrast compared to the surrounding smaller LPSCl particles, and well within the expected range for these unmodified particle sizes. Smaller LPSCl particles are surrounding the sulfur particle, where their size is also retained. It is possible that some particles that experienced high enough frictional stress were reduced slightly after the 1-hour high energy milling process. However, this reduction if any, was not significant enough given the ability to still identify large sulfur particles, nor was it significant enough to improve the Bulk S composite's rate performance, requiring further particle size reduction. For the micron and sub-micron case, particle sizes were not significantly reduced. We also conducted SEM-EDS mapping on the Micron and Sub-micron Sulfur composites, where particles were found to still be on the micron scale.

We included a new Supplementary Information figure to support the need and motivation of pre-milling sulfur for enabling higher rate performance, shown below.

New Fig. S24

Fig S24. SEM-EDS mapping of the cathode composite powders synthesized with Bulk Sulfur.

To address the potential changes of crystallinity of sulfur after the pre-milling step, we conducted XRD on the milled sulfur samples, comparing to the unmodified Bulk sulfur. The spectra are shown below. All sulfur samples exhibit diffraction patterns consistent with elemental sulfur, with no observable changes in crystallinity. The main diffraction peak at the (222) reflection is present at $10.5^\circ 2\theta$ for all samples, indicating no changes within the lattice. These results validate that pre-milling sulfur to reduce its particle size does not affect its crystallinity or structure and only reduces the size of their particles. The XRD spectra has also been added into the Supplementary Information.

New fig. S26.

Fig. S26. XRD spectra of sulfur particles after particle size reduction to the micron and sub-micron scale.

The following discussion has been added to the manuscript, referencing the new Supplementary figures. “After composite synthesis, which subjects sulfur to high energy milling with LPSCI and carbon, it is possible that particle sizes can reduce, although, sulfur particles were found to retain close to their original size as observed from the SEM-EDS mapping results (**fig. S24**). These particle sizes, while sufficient for low-rate cycling, need to be reduced to improve Li^+ transport and enable higher rate operation. Therefore, sulfur particle sizes were produced on the micron and sub-micron scale (**fig. S25**). XRD was conducted after the size reduction process to ensure their crystallinity was preserved (**fig. S26**).”

Comment 5: Clarification of cell compositions used in electrochemical tests. Due to the large number of cathode composite configurations and corresponding electrochemical data, it is difficult to follow the overall flow. Please clearly summarize and specify the cell configurations used in each data panel to improve readability and interpretation.

Response: We appreciate the reviewer’s suggestion and provided extra clarification to follow the experimental workflow. We have made sure to specify each cell configuration within the presented electrochemical results. These changes have been reflected within the main manuscript and the Supplementary Information when necessary. We have also added the cell configurations in the figures where multiple cell configurations were used.

When introducing Fig. 2a, we have included details of the cathode composition and cell configuration within the manuscript.

“A commonly used sulfide electrolyte, carbon type, and weight composition was employed for our study, where unmodified elemental sulfur, argyrodite LPSCI, and acetylene black (AB) carbon were used to fabricate a cathode composite with a weight ratio of 30:50:20 wt.%. This composition was used in all electrochemical tests unless otherwise stated.”

*“To compare the different cathode synthesis methods, cells were fabricated with LPSCI as the separator layer and a Li-In anode. **Figure 2a** shows their first cycle voltage profiles where increases in utilization and capacity are attained by introducing high energy milling steps to the fabrication process.”*

For Fig. S8, we have added the following within the manuscript.

“The interaction between LPSCI and Li_2S was also investigated. A comparison between cathode preparation methods was conducted, where half cells were fabricated with a Li-In anode.” We have also clarified the cell configuration within the figure.

For Fig. 3, we have added the cell composition within the figure caption.

*“**Fig. 3 | LPSCI redox activity and capacity contribution.** Voltage profiles of LPSCI/C Li-In half cells evaluated under (a) sulfur voltage limits. b, Cyclic voltammetry of sulfur and LPSCI cathode composites with a Li-In anode. c, dQ/dV curves of the 1st and (d) 3rd cycle for cells with decreasing sulfur wt.% (S/LPSCI/C). The areal loading of sulfur was 0.3 mg cm^{-2} , 0.84 mg cm^{-2} , and 1 mg cm^{-2} corresponding to a total active mass of 2.8 mg cm^{-2} , 2.9 mg cm^{-2} , and 2.5 mg cm^{-2} for 10 wt.%, 20 wt.%, and 30 wt.% sulfur composites, respectively.”*

For Fig. 5d, e, and f, we introduced the experimental validation results by adding the following text within the manuscript.

“To validate the modeling results, room temperature electrochemical performance was performed evaluating each sulfur particle class in Li-In half cells.”

For Fig. 6, we have added the cell configuration information within the Fig. 6c and 6d.

Fig. 6 | Quantifying cathode and cell level volume changes. Cryo-FIB images of the (a) micron sulfur and (b) Li_2S cathode composite at various states of charge. c, *Operando* pressure monitoring of LCO and Sulfur using a lithiated silicon anode and of (d) LCO and Li_2S using a μSi anode during the first formation cycle.

Reviewer #5

In this work, the authors report the fabrication of high-performance all-solid-state sulfur/ Li_2S cathodes using a one-step high-energy ball milling method. This process induces the formation of sulfur-rich thiophosphate phases (Li_3PS_4+n), which facilitate enhanced interfacial ion transport. The study presents a comprehensive investigation into the structural evolution of cathode materials during cycling, the redox activity of cathode components, the influence of electrode microstructure (particularly sulfur particle size) on battery performance, volume-pressure evolution during electrochemical cycling, and the overall electrochemical performance of the all-solid-state

batteries. The performance demonstrated in both half-cells and practical full pouch cells is promising and offers insights into the development of practical solid-state lithium-sulfur batteries.

Comment 1: The areal sulfur loading should be clearly specified for all electrochemical data, particularly in Fig. 2a, Fig. 3, and Fig. 5e.

Response: The areal sulfur loading has been added to the electrochemical data presented in the manuscript. These changes are shown for each figure below.

Fig. 2 | Characterizing the sulfur cathode composites after synthesis. **a**, Voltage profiles of the investigated cathode composite preparation methods in Li-In half cells. **b**, X-ray diffraction (XRD) spectra of the cathode composite after fabrication. **c**, Raman spectroscopy and **(d)** X-ray absorption spectroscopy (XAS) of the one-step milled composite. **e**, Schematic illustrating the surface reaction with LPSCI bonded on the sulfur surface as a result of the one-step milling procedure.

Fig. 3 | LPSCI redox activity and capacity contribution. Voltage profiles of LPSCI/C Li-In half cells evaluated under (a) sulfur voltage limits. b, Cyclic voltammetry of sulfur and LPSCI cathode composites with a Li-In anode. c, dQ/dV curves of the 1st and (d) 3rd cycle for cells with decreasing sulfur wt.% (S/LPSCI/C). The areal loading of sulfur was 0.3 mg cm^{-2} , 0.84 mg cm^{-2} , and 1 mg cm^{-2} corresponding to a total active mass of 2.8 mg cm^{-2} , 2.9 mg cm^{-2} , and 2.5 mg cm^{-2} for 10 wt.%, 20 wt.%, and 30 wt.% sulfur composites, respectively.

Fig. 5 | Modeling sulfur cathode microstructures with electrochemical validation. **a**, Geometrically modeled sulfur composite electrodes for bulk, micron, and sub-micron sulfur particles. **b**, Specific active surface area and **(c)** ionic transport tortuosity as a function of AM content (%) and particle size. **d**, First formation cycle voltage profiles of sulfur composites with bulk, micron, and sub-micron particles at C/20 ($1C = 1.6 \text{ mA cm}^{-2}$). **e**, Long-term cycling stability at C/2 with discharge capacity being normalized by the active mass (sulfur and LPSCI). **f**, Nyquist plots and equivalent circuit fitting results from EIS measurements of the micron (top) and sub-micron (bottom) sulfur cathodes at cycle 100, 300, and 500 in their de-lithiated states. **g**, Distribution of the von Mises stress on the SSE matrix after simulated lithiation. **h**, Predicted variation of maximum von Mises stress.

Comment 2: For improved clarity, the authors should provide the calculated ionic conductivity values of the cathode composites and Li6PS5Cl (LPSCI) prepared under different milling conditions, in addition to the EIS spectra shown in Fig. S3c,f.

Response: We appreciate the reviewer's suggestion and have added the calculated ionic conductivity values within the manuscript for clarity.

“Ionic conductivity of the S/LPSCl composites after increasing milling durations were also measured, where an increase from $6 \times 10^{-6} \text{ S cm}^{-1}$ to $2 \times 10^{-5} \text{ S cm}^{-1}$ is observed after 1 hour and saturates with continued milling to 10-hours (Fig. S4c). Standalone XRD, Raman, and EIS was also measured for the LPSCl catholyte at 1- and 10-hour milling intervals (Fig. S4d-f). Contrary to what was observed for the S/LPSCl composites, continued milling of LPSCl results in a decrease in ionic conductivity from $1.3 \times 10^{-3} \text{ S cm}^{-1}$ to $4 \times 10^{-5} \text{ S cm}^{-1}$, attributed to the formation of insulative Li_2S and LPS phases.”

These calculated ionic conductivities have also been added to Fig. S3c and S3f.

Fig. S4. Characterization of sulfur and LPSCl catholyte after one-step synthesis without carbon. **a**, XRD spectra of S/LPSCl composite with increased milling durations. **b**, Raman spectra of composites shown in **a**. **c**, Corresponding Nyquist plots of S/LPSCl composites with increasing milling durations. Standalone **d**, XRD spectra **e**, Raman spectra, and **f**, Nyquist plots of LPSCl after 1 hour and 10 hour milling durations with their calculated ionic conductivity.

Comment 3: As X-ray diffraction (XRD) patterns only indicate the crystal structure of the materials, they are unable to identify the amorphous sulfur-rich interlayer (Li_3PS_4+n) after cycling (line 331). Instead, Raman spectroscopy should be conducted to validate the existence of the interlayer phase after cycling.

Response: The reviewer has provided an insightful suggestion. XRD provided structural information confirming that the poorly crystalline sulfur composite is retained. To support the

assumption that the amorphous sulfur-rich interlayer is preserved, Raman was conducted on sulfur cathodes that were cycled for 5 and 100 cycles. This was done to validate the long-term existence of the formed conductive interfaces.

The Raman spectra of the cycled cathodes are shown below where the $\text{Li}_3\text{PS}_{4+n}$ phase was identified after short and long-term cycling. A clear redshift is observed in the spectra at the symmetric S-S bending region, from 153 cm^{-2} to 150 cm^{-1} . These results are comparable to the spectra of the cathode composite after synthesis, exhibiting a peak attributed to the S-S bending region at 149.5 cm^{-1} . The retainment of this conductive phase helps explain the long cycle life and sustained sulfur utilization obtained with this system.

The following figure has been added to the Supplementary Information as Fig. S14.

New Fig. S14

Fig. S14. Raman spectra focusing on the symmetric S-S bending region of sulfur cathodes after cycling.

The following discussion has been added to the manuscript.

“The high charge capacity yet low peak intensity and amorphous background indicate that the reformed sulfur is amorphous as in the pristine state and suggests the sulfur rich interlayer is preserved with cycling. This was confirmed by conducting Raman spectroscopy on the cycled sulfur cathode, where a redshift to lower wavenumbers at the S-S bending region is observed (fig. S14), as previously shown for the pristine composite.”

Comment 4: Although sulfur with different particle sizes was used, the high-energy ball milling process shall further reduce their particle sizes. Therefore, SEM-EDS mapping of the pristine cathode surface is recommended to examine the actual sulfur particle size after fabrication.

Response: The reviewer brings up a valid point as we were also curious about the possibility of reducing particle sizes with the planetary ball mill during the short (1 hour) high energy (500 rpm) milling protocol. SEM was first conducted on the unmodified sulfur composite after

synthesis. We broke up the previous Fig. S16 to help with the flow for the addition of SEM-EDS mapping shown below. SEM of the unmodified Bulk sulfur powder is shown in Fig. S22a, where a wide particle size distribution from 100 microns to 10 microns is observed. After the high energy ball-milling process, a large sulfur particle is identified surrounded by smaller LPSCI particles shown in Fig. S22b. This distinction was made due to their differences in contrast (differences in atomic number between sulfur and LPSCI) from the backscattered electron detector. This large sulfur particle is near 50 μm in diameter, exhibiting a comparable size prior to composite synthesis.

Fig. S22. Scanning electron microscopy (SEM) images of unmodified sulfur and composite. Powders of a) Bulk Sulfur as received and b) composite after synthesis. c) Top view after cathode composite fabrication.

SEM-EDS mapping was also conducted on another batch of composite to verify. We did find EDS mapping to be challenging on the composite since LPSCI will contribute to the sulfur signal. The Cl and P K mapping was also included but was still hard to discern the particles due to the good composite distribution and interfacial bonding between sulfur and the SSE. However, the P and Cl mapping do show lower intensity at the location of the highlighted sulfur particle. These results are shown below.

A large sulfur particle, exhibiting a much darker contrast compared to the surrounding smaller LPSCI particles was identified and labeled in the above image. Its particle size is 30 μm in diameter, still on the expected range for precursor sulfur particle size and also indicating that particles still need to be further reduced. Smaller LPSCI particles surrounding the sulfur particle, shows that its size is also retained. It is possible that some particles that experienced high enough frictional stress were reduced slightly after the 1-hour high energy milling process. However, this reduction if any, was not significant enough given the ability to still identify large sulfur particles, nor was it significant enough to improve the Bulk S composite's rate performance, requiring further particle size reduction.

For the micron and sub-micron case, particle sizes were found to not significantly reduced either.

Fig. S25. Scanning electron microscopy (SEM) images of milled sulfur particles and their composites. Powders of a) Micron scale sulfur, b) composite after synthesis, and c) top view after cathode composite fabrication. Powders of d) Sub-micron scale sulfur, e) composite after synthesis, and f) top view after cathode composite fabrication.

We also conducted SEM-EDS mapping on the Micron and Sub-micron Sulfur composites, where particles were found to still be on the micron scale.

We included a new Supplementary Information figure to support the need and motivation of pre-milling sulfur to enable high rate performance, shown below.

New Fig. S24

Fig S24. SEM-EDS mapping of the cathode composite powders synthesized with Bulk Sulfur.

As suggested by the reviewer, we also did SEM-EDS mapping of the cathode composite after fabrication. A slightly lower densification time was used for these samples in an effort to better differentiate sulfur and LPSCl. The higher contrast of these images compared to those in Fig. S22 are due to the higher accelerating voltage of 10kV used here for EDS acquisition. This higher accelerating voltage better highlights interfaces between the densified particles. While we are unable to fully differentiate the particles using EDS, the SEM images of the powder composites provide enough evidence to show that the sulfur and LPSCl particles are not drastically reduced after the short duration high energy milling. Pre-milling the sulfur particles prior to composite synthesis is helpful to improve interfacial contact between all components and create more surface area for sulfur and LPSCl to interracially react.

The following discussion has been added to the manuscript, referencing the new Supplementary figures to validate that particles are not significantly reduced after synthesis.

“After composite synthesis, which subjects sulfur to high energy milling with LPSCI and carbon, it is possible that particle sizes can reduce, although, sulfur particles were found to retain close to their original size as observed from the SEM-EDS mapping results (fig. S24). These particle sizes, while sufficient for low rate cycling, need to be reduced to improve Li^+ transport and enable higher rate operation. Therefore, sulfur particle sizes were produced on the micron and sub-micron scale (fig. S25). XRD was conducted after the size reduction process to ensure their crystallinity was preserved (fig. S26).”

Comment 5: It is unclear in Fig. 6c and d which curves correspond to pressure changes, and which represent voltage during cycling. The figure should be redrawn to improve clarity. Additionally, due to the relatively small pressure changes in all-solid-state lithium-sulfur batteries, the authors are encouraged to reduce the y-axis range for pressure to better illustrate the pressure evolution trends during cycling.

Response: We appreciate the reviewer’s suggestion and have made sure to differentiate between the pressure and voltage profile curves by adding their own y-axis in a stacked subplot format. The y-axis range for the pressure curves was also reduced to better illustrate the pressure changes with cycling. These changes to Fig. 6c and 6d are shown below.

Fig. 6 | Quantifying cathode and cell level volume changes. Cryo-FIB images of the (a) micron sulfur and (b) Li₂S cathode composite at various states of charge. c, *Operando* pressure monitoring of LCO and Sulfur using a lithiated silicon anode and of (d) LCO and Li₂S using a μSi anode during the first formation cycle.

Comment 6: It is recommended to use the same color (yellow) for sulfur particles in fig. 7d as in Fig. 7a to maintain consistency.

Response: We thank the reviewer for their suggestion and have changed the sulfur particle color to yellow in Fig.7d for consistency. This modification is shown below.

Fig. 7 | High loading sulfur and Li₂S electrochemical performance with energy density outlook. **a**, Schematic of the sulfur cathode half-cell architecture in pellet cells. **b**, First cycle voltage profiles with increasing areal capacity. **b**, Long term cycling stability at the 11 mAh cm⁻² level evaluated at room temperature. **d**, Schematic of the dry process sulfur cathode paired with a lithiated silicon anode in pellet cells. **e**, Rate performance at the 5.5 mAh cm⁻² level. **f**, Cycling stability at the 7.4 mAh cm⁻² level under C/5 current density. **g**, Theoretical gravimetric energy density as a function of wt.% of sulfur and areal capacity. **h**, Gravimetric energy density comparison between sulfur and Li₂S cathodes with various anodes. Values used for these calculations can be found in table S7. **i**, Schematic of the Li₂S/anode-free pouch cell. **j**, First formation cycle at C/10. **k**, Cycling performance at the 4.5 mAh cm⁻² level under 10MPa and 60°C conditions.

Comment 7: The current densities are reported in C-rate, but it is unclear how 1C is defined. Given the varying cathode compositions (e.g., sulfur, Li₂S, and LPSCI), all current densities are encouraged to be reported in A g⁻¹ based on the total weight of sulfur/Li₂S and LPSCI.

Response: We thank the reviewer for their careful review and suggestion. Based on their feedback, we have reported current densities in $A\ g^{-1}$ for all electrochemical data presented in the main figures of the manuscript. These changes are shown below for each modified figure.

Fig. 2 | Characterizing the sulfur cathode composites after synthesis. **a**, Voltage profiles and **(b)** X-ray diffraction (XRD) spectra of the investigated cathode preparation methods. Cells were constructed at the $1.6\ mAh\ cm^{-2}$ level. **c**, Raman spectroscopy and **(d)** X-ray absorption spectroscopy (XAS) of the one-step milled composite. **e**, Schematic illustrating the surface reaction with LPSCI bonded on the sulfur surface as a result of the one-step milling procedure.

Fig. 3 | LPSCI redox activity and capacity contribution. Voltage profiles of LPSCI/C Li-In half cells evaluated under (a) sulfur voltage limits. b, Cyclic voltammetry of sulfur and LPSCI cathode composites with a Li-In anode. c, dQ/dV curves of the 1st and (d) 3rd cycle for cells with decreasing sulfur wt.% (S/LPSCI/C). The areal loading of sulfur was 0.3 mg cm^{-2} , 0.84 mg cm^{-2} , and 1 mg cm^{-2} corresponding to a total active mass of 2.8 mg cm^{-2} , 2.9 mg cm^{-2} , and 2.5 mg cm^{-2} for 10 wt.%, 20 wt.%, and 30 wt.% sulfur composites, respectively. Current densities used in the dQ/dV curves were 0.06 A gs^{-1} , 0.07 A gs^{-1} , and 0.08 A gs^{-1} for 10 wt.%, 20 wt.%, and 30 wt.% sulfur composites, respectively.

Fig. 5 | Modeling sulfur cathode microstructures with electrochemical validation. **a**, Geometrically modeled sulfur composite electrodes for bulk, micron, and sub-micron sulfur particles. **b**, Specific active surface area and **(c)** ionic transport tortuosity as a function of AM content (%) and particle size. **d**, First cycle voltage profiles at $C/20$ ($1\text{C} = 1.6\text{ mA cm}^{-2}$) **e**, Long-term cycling stability at $C/2$ with discharge capacity being normalized by the active mass (sulfur and LPSCI). **f**, Nyquist plots and equivalent circuit fitting results from EIS measurements of the micron (top) and sub-micron (bottom) sulfur cathodes at cycle 100, 300, and 500 in their delithiated states. **g**, Distribution of the von Mises stress on the SSE matrix after simulated lithiation. **h**, Predicted variation of maximum von Mises stress.

Fig. 7 | High loading sulfur and Li₂S electrochemical performance with energy density outlook. **a**, Schematic of the sulfur cathode half-cell architecture in pellet cells. **b**, First cycle voltage profiles with increasing areal capacity. **b**, Long term cycling stability at the 11 mAh cm^{-2} level evaluated at room temperature. **d**, Schematic of the dry process sulfur cathode paired with a lithiated silicon anode in pellet cells. **e**, Rate performance at the 5.5 mAh cm^{-2} level. **f**, Cycling stability at the 7.4 mAh cm^{-2} level under C/5 current density. **g**, Theoretical gravimetric energy density as a function of wt.% of sulfur and areal capacity. **h**, Gravimetric energy density comparison between sulfur and Li₂S cathodes with various anodes. Values used for these calculations can be found in table S7. **i**, Schematic of the Li₂S/anode-free pouch cell. **j**, First formation cycle at C/10. **k**, Cycling performance at the 4.5 mAh cm^{-2} level under 10MPa and 60°C conditions.

The following Supplementary Information figures, Fig. S7, Fig. S8, Fig S10, Fig. S11, Fig. S18, Fig. S19, Fig. S24 have also been updated to report current densities in A g^{-1} .

Comment 8: The authors claim the formation of both Li_3PS_4 and $\text{Li}_4\text{P}_2\text{S}_6$ based on a single XRD crystalline peak (Fig. 4b). This is not sufficient evidence. Additional techniques such as solid-state NMR or Raman spectroscopy should be employed to verify the presence of $\text{Li}_4\text{P}_2\text{S}_6$. Moreover, prior literature reports $\text{Li}_4\text{P}_2\text{S}_7$ rather than $\text{Li}_4\text{P}_2\text{S}_6$ as an oxidation product of LPSCl (Nature Materials 2020, 19, 428–435.) and this discrepancy should be addressed.

Response: We thank the reviewer for giving us the opportunity to validate and clarify our findings. We also agree that additional characterization methods should be employed to verify the formation of Li_3PS_4 and $\text{Li}_4\text{P}_2\text{S}_6$ after cycling. Therefore, we conducted Raman spectroscopy on the cycled sulfur cathodes. Raman spectroscopy is a local technique with a lateral resolution of 1 μm . Given that the micron sulfur cathode composite consists of a distribution of elemental sulfur w/ sulfur rich $\text{Li}_3\text{PS}_{4+n}$, and LPSCl particles ranging from 1-10 microns, various locations need to be measured to elucidate the entire cathode composition.

Previously, spectra were collected at a region consisting of sulfur with evidence of the sulfur rich $\text{Li}_3\text{PS}_{4+n}$ phase. This was shown in the new fig. S14 and addressed in Comment 3, providing evidence for the retainment of the conductive interlayer with cycling. From the XANES and XRD results, we expected LPSCl to be retained with some LPS formed after cycling. These results were validated by the newly collected Raman spectra shown below. At regions correlated to be within the SSE matrix of the cathode composite, we were able to identify both $\text{Li}_6\text{PS}_5\text{Cl}$ (LPSCl) and Li_3PS_4 (LPS), with their signals convoluted in one peak.

New Fig. S15

Fig. S15. Raman spectra of the cycled sulfur cathode at a region correlating within the SSE bulk. a) Comparison to pristine LPSCl and b) fitted spectra highlighting two different PS_4^{3-} phases attributed to LPSCl and LPS.

Curve fitting was performed to deconvolute the measurement, revealing two peaks at 425 cm^{-1} and 418 cm^{-1} both attributed to the thiophosphate unit PS_4^{3-} present in LPSCl and LPS. We also added the reference spectra of LPSCl, which highlights that LPSCl only exhibits one main peak at 425 cm^{-1} .

The reviewer is correct that $\text{Li}_4\text{P}_2\text{S}_7$ is a primary decomposition product (at the charged state) of LPSCl and is discussed in multiple works. If $\text{Li}_4\text{P}_2\text{S}_7$ or $\text{Li}_4\text{P}_2\text{S}_6$ was present, a peak would be

identified at 400 cm^{-1} and 385 cm^{-1} in the Raman spectra (R. Xu et al., *J. Mater. Chem. A*, 2017,5, 2829-2834). However, we were also unable to confirm the presence of either $\text{Li}_4\text{P}_2\text{S}_7$ or $\text{Li}_4\text{P}_2\text{S}_6$ using Raman. In the above spectra, there is a small hump observed near 385 cm^{-1} , which is the expected peak position of $\text{P}_2\text{S}_6^{4-}$ but its intensity is very low and would be limited evidence to claim $\text{P}_2\text{S}_6^{4-}$ being formed. Therefore, we have decided to not claim $\text{Li}_4\text{P}_2\text{S}_6$ formation in the manuscript and attribute the peak present in the XRD results of the cycled cathode in Fig. 4b to γ -LPS, validated by Raman spectroscopy and XAS. Below shows the updated Fig. 4 in the main manuscript.

Fig. 4 | Probing electrochemical reversibility. a, Nyquist plots of *in-situ* EIS measurements of the sulfur cathode composite during the first formation cycle. b, XRD and (c) TGA of high loading (7.6 mg cm^{-2}) sulfur cathodes at the pristine, discharged, and charged states. Quantified sulfur masses are listed in *table S4*. XANES fitting results as weight percentages of products at each state of charge for (d) LPSCI/C (e) sulfur/LPSCI/C, and (f) Li_2S /LPSCI/C composites. Error bars represent the uncertainty of the quantified masses. The electrochemical performance of cells used for XRD and XANES are presented in **Fig. S20** and **S21**.

We have also added the following discussion referencing the new Supplementary Information figure Fig. S15, confirming the presence of LPS after cycling.

“A new diffraction peak at $15^\circ 2\theta$ is also identified after the 1st cycle. Raman spectra collected at a region within the SSE bulk shows a convoluted PS_4^{3-} peak, attributed to a mixture of LPSCI and LPS (**fig. S15**). This new phase after the 1st cycle can be attributed to γ -LPS, validating the active

participation and reversibility of LPS, as hypothesized from the electrochemical and dQ/dV results.”

Comment 9: The error bars in the figures should be clearly defined. The number of replicates used to generate the error bars should be specified in the respective figure captions.

Response: The error bars are shown in the LCF results in Fig. 4d-f. These represent the standard error and uncertainty for that specific component weight calculation. The uncertainty is computed during non-linear least squares minimization (performed with Athena Software) between the measured spectra and the linear combination of the reference spectra. The covariance matrix provides the statistical relationships of the fitted parameters as the variance for each computed weight. The standard deviation or error for each weight percentage is then computed by taking the square root of the variance. Weights with high uncertainty can be due to noise or if reference spectra possess similar features. We have defined the error bars in the Fig. 4d, e, f caption shown below.

“Fig. 4 | Probing electrochemical reversibility. a, Nyquist plots of in-situ EIS measurements of the sulfur cathode composite during the first formation cycle. **b,** XRD and (c) TGA of high loading (7.6 mg cm^{-2}) sulfur cathodes at the pristine, discharged, and charged states. Quantified sulfur masses are listed in table S4. XANES fitting results as weight percentages of products at each state of charge for (d) LPSCI/C (e) sulfur/LPSCI/C, and (f) $\text{Li}_2\text{S/LPSCI/C}$ composites. Error bars represent the standard error and uncertainty of the quantified masses. The electrochemical performance of cells used for XRD and XANES are presented in Fig. S20 and S21.”

Comment 10: The figure caption for Fig. 5f should explicitly state whether the EIS spectra were collected at the charged or discharged state.

Response: The reviewer has brought up a good point. We have added the lithiation state of the sulfur cathodes. All cells were in the charged (de-lithiated) state at their stated cycle life when EIS was conducted. This modification is shown in the Fig. 5f caption below.

“Fig. 5 | Modeling sulfur cathode microstructures with electrochemical validation in Li-In half cells. a, Geometrically modeled sulfur composite electrodes for bulk, micron, and sub-micron sulfur particles. **b,** Specific active surface area and (c) ionic transport tortuosity as a function of AM content (%) and particle size. **d,** First cycle voltage profiles of cathode composites with various particles sizes **e,** Long-term cycling stability at C/2 with discharge capacity being normalized by the active mass (sulfur and LPSCI). **f,** Nyquist plots and equivalent circuit fitting results from EIS measurements of the micron (top) and sub-micron (bottom) sulfur cathodes at cycle 100, 300, and 500 in their de-lithiated states. **g,** Distribution of the von Mises stress on the SSE matrix after simulated lithiation. **h,** Predicted variation of maximum von Mises stress.”

Comment 11: On line 354, what is L4.1PS4 phase? Is it Li4PS4 or Li4.1PS4 phase?

Response: Within the sulfur voltage window of 1-3V, LPSCI will not fully reduce. We hypothesize the formation of a lithium rich Li_4PS_4 phase, Li_2S , and LiCl . The incomplete reduction of the SSE preserves the composite's ionic conductivity for reversible electrochemical performance. We verify LPS formation within the LPSCI/C cathode cell using XAS. We also verify the presence of LPS within the sulfur system using XRD, Raman, and XAS.

Comment 12: Regarding the questions raised by previous Reviewer #2, the authors have addressed some of them but the following still needs further elucidation: Comment 3 and 8 by Reviewer #2 The authors claim the formation of $\text{Li}_4\text{P}_2\text{S}_6$, but as previously noted, this has not been conclusively demonstrated. Additional characterization is needed to confirm its presence.

Response: The reviewer brings up a good point which was addressed earlier in Comment 8. We conducted Raman spectroscopy on the cycled sulfur cathodes. At a region correlated to be within the SSE bulk of the composite with no observable signs of sulfur, we were able to identify both LPSCI and LPS at 425 cm^{-1} and 418 cm^{-1} . These new results are presented in Fig. S15. A small hump was also observed near 385 cm^{-1} , which is the expected location for $\text{P}_2\text{S}_6^{4-}$, but is limited evidence to claim its formation. Therefore, we have removed the claim of forming $\text{Li}_4\text{P}_2\text{S}_6$ within the manuscript and attribute the XRD peak of the cycled cathode composite to be from γ -LPS only. This is consistent to where the main diffraction peak of γ -LPS should be located. The formation of LPS was also verified using Raman spectroscopy, clearly shown from the peak fitting results in Fig. S15b. The newly collected Raman spectra is shown below.

New Fig. S15.

Fig. S15. Raman spectra of the cycled sulfur cathode at a region correlating within the SSE bulk. a) Comparison to pristine LPSCI and b) fitted spectra highlighting two different PS_4^{3-} phases attributed to LPSCI and LPS.

The discussion in the manuscript has been revised to the following.

“A new diffraction peak at $15^\circ 2\theta$ is also identified after the 1st cycle. Raman spectra collected at a region within the SSE bulk shows a convoluted PS_4^{3-} peak, attributed to a mixture of LPSCI and LPS (fig. S15). This new phase after the 1st cycle can be attributed to γ -LPS, validating the active participation and reversibility of LPS, as hypothesized from the electrochemical and dQ/dV

results. The differences between the pristine and charged composite mass loss originate from the sulfur rich interlayer, which increases the thermal stability of the reacted sulfur. The preservation of ionically conductive interfaces along with SSE redox helps sustain high sulfur utilization and cycle life.”

Comment 13: Comment 6 by Reviewer #2 - Redox Reactions. In the response, the authors state that the cathode contains 43.5 wt% LPSCl, 6.5 wt% $\text{Li}_3\text{PS}_{4+n}$, and 23.5 wt% sulfur, and that the capacity contribution from $\text{Li}_3\text{PS}_{4+n}$ is marginal. However, this analysis is flawed. First, if 6.5 wt% of sulfur has reacted, the corresponding $\text{Li}_3\text{PS}_{4+n}$ must be more than 6.5 wt% due to the added mass of the thiophosphate framework. Second, the chain length of sulfur in $\text{Li}_3\text{PS}_{4+n}$ is unknown, making accurate quantification via TGA alone infeasible. Third, the theoretical capacity of $\text{Li}_3\text{PS}_{4+n}$ (depending on the sulfur chain length) can exceed 500 mAh g⁻¹, making its capacity contribution significant (see Angew. Chem. Int. Ed., 2013, 52, 7608–7611).

Response: We thank the reviewer for bringing up this point. The reviewer is correct that we are unable to accurately quantify the sulfur chain length of this phase using TGA. Instead, we can use the TGA results to quantify the unreacted and reacted sulfur mass within the composite. This allows us to estimate the expected capacity we would obtain using a model cell composition and compare with our electrochemical results. We can also use the dQ/dV curves of cells that consist of different sulfur wt.% to determine the reduction potentials of the $\text{Li}_3\text{PS}_{4+n}$ phase and LPSCl, where the capacity obtained at those reduction potentials can be correlated back from our electrochemical results.

Using a model cell composition of 1.6 mAh cm⁻² which correlates to 2.5mg of composite, we can estimate the capacity of unreacted and reacted sulfur assuming full utilization (1675 mAh g⁻¹) and LPSCl capacity based on the constant current discharge and charge results. A total discharge and charge capacity of 1.38 mAh and 1.70 mAh is expected. This is quite close to our electrochemical results, where a cell with this loading delivers near 1.2 mAh and 1.6 mAh during the discharge and charge. Since we know the reacted sulfur exists in the $\text{Li}_3\text{PS}_{4+n}$ phase, its full utilization is not likely. Subtracting the unreacted sulfur mass capacity, estimated to be 1 mAh, from the total discharge capacity, we can assign the remaining 0.20 mAh to $\text{Li}_3\text{PS}_{4+n}$ and LPSCl.

From the dQ/dV curves, we expect the reacted sulfur in $\text{Li}_3\text{PS}_{4+n}$ to reduce first and follow similar redox behavior as LPS. This reduction peak is observed for all cells but is prominent for the cell with the lowest sulfur wt.% since the reduction of $\text{Li}_3\text{PS}_{4+n}$ dominates the reduction capacity.

We can see that the LPSCI/C cell clearly reduces after $\text{Li}_3\text{PS}_{4+n}$. $\text{Li}_3\text{PS}_{4+n}$ phase reduces at 1.3V while LPSCI reduces at 1.15V. Using these reduction potentials from the dQ/dV curves, we can correlate the obtained capacity from our electrochemical results. From OCV to 1.3V, 1 mAh is obtained as estimated above. The remaining capacity from 1.3 to 1.15 is 0.09 mAh. From 1.15 to 1V (our lower cutoff), 0.11 mAh capacity is obtained, attributed to LPSCI.

Capacity Estimated based on potentials from dQ/dV				
	Unreacted Sulfur (23.5 wt.%) (mAh)	Reacted Sulfur in $\text{Li}_3\text{PS}_{4+n}$ (6.5 wt.%) (mAh)	LPSCI (50 wt.%) (mAh)	Total Capacity (mAh)
Discharge	1	0.09	0.11	1.20
Charge	1	0.16	0.44	1.60

The specific capacity of sulfur in $\text{Li}_3\text{PS}_{4+n}$ is estimated to be 553 mAh g^{-1} during the discharge. This suggests $\text{Li}_3\text{PS}_{4+n}$ exists as $\text{Li}_3\text{PS}_{4+3}$ after synthesis. This aligns well with the large Raman redshift, which correlates to shorter sulfur chains.

$$Q = \frac{nF}{3.6M}$$

$$n = 6, F = 96485 \left[\frac{\text{C}}{\text{mol}} \right], M = \text{molecular mass} \left[\frac{\text{g}}{\text{mol}} \right]$$

$$Q_{\text{red}}/\text{Capacity} = 582 \text{ mAh g}^{-1}$$

Since we validated the existence of this phase after cycling, we can conclude that its redox is reversible although a higher capacity is expected during the charging process. Using these estimations, we can assign capacity contributions for each component.

Capacity Contribution (%)				
	Unreacted Sulfur (23.5 wt.%) (mAh)	Reacted Sulfur in $\text{Li}_3\text{PS}_{4+n}$ (6.5 wt.%) (mAh)	LPSCI (50 wt.%) (mAh)	Total Capacity (mAh)
Discharge	83.3	7.5	9.2	1.20
Charge	62.1	10	27.5	1.60

Table S2. Estimated capacity contributions between unreacted sulfur, reacted sulfur in $\text{Li}_3\text{PS}_{4+n}$, and LPSCI.

Expected Capacity Based on Mass for each Component				
	Unreacted Sulfur (mAh)	Reacted Sulfur in $\text{Li}_3\text{PS}_{4+n}$ (mAh)	LPSCI (mAh)	Total Capacity (mAh)
Discharge	1	0.25	0.13	1.38
Charge	1	0.25	0.44	1.70
Capacity Estimated based on Electrochemistry				
Discharge	1	0.20		1.20
Charge	1	0.16	0.44	1.60
Capacity Estimated based on potentials from dQ/dV				
Discharge	1	0.09	0.11	1.20
Charge	1	0.16	0.44	1.60
Capacity Contributions (%)				
Discharge	83.3	7.5	9.2	1.20
Charge	62.1	10	27.5	1.60

These estimations have been added to the supplementary information as Table S2, with additional discussion regarding the capacity contribution of the reacted sulfur in the $\text{Li}_3\text{PS}_{4+n}$ phase.

“In this system, three active materials exist: bulk LPSCI, the formed interlayer phase ($\text{Li}_3\text{PS}_{4+n}$) consisting of sulfur that has reacted with LPSCI, and unreacted elemental sulfur. TGA was used to quantify the reacted and unreacted sulfur mass, where 23.5 out of the 30 wt.% of sulfur remains unaltered after synthesis. Using the known sulfur and LPSCI mass, we can estimate the capacity contributions of the active materials and compare to what was obtained experimentally. For a 1.6 mAh cm^{-2} cell using a total composite mass of 2.5 mg, 23.5 wt.% unreacted sulfur is expected to deliver 1 mAh (using 1675 mAh g^{-1}). Therefore, during the discharge, unreacted bulk

sulfur is estimated to contribute 83% of the capacity with the remainder attributed to LPSCl and $\text{Li}_3\text{PS}_{4+n}$ reduction. To isolate the capacity expected from just LPSCl, cells with just LPSCl and carbon were fabricated using a Li-In alloy as a lithium source.

“Subtracting the expected LPSCl and unreacted bulk sulfur capacity from the total cell capacity obtained after the charge leaves the remaining capacity to be assigned to reacted sulfur in $\text{Li}_3\text{PS}_{4+n}$. This equates to a capacity contribution of 10% for the reacted sulfur and 27.5% for LPSCl after the charge, suggesting that the reacted sulfur in $\text{Li}_3\text{PS}_{4+n}$ and LPSCl is redox-active and actively participates capacity during cycling.”

“We can estimate their individual discharge capacity contributions based on the reduction potentials obtained from dQ/dV . From OCV to 1.3V, 1 mAh is obtained, as estimated above. The remaining capacity from 1.3 to 1.15V can be attributed to $\text{Li}_3\text{PS}_{4+n}$, which is responsible for 7.5% of the overall discharge capacity. The remaining capacity obtained from 1.15 to 1V is from LPSCl, which contributes 9.2%. The specific capacity of the reacted sulfur is estimated to be 553 mAh g^{-1} during the discharge, likely existing in the $\text{Li}_3\text{PS}_{4+3}$ phase after synthesis. These calculations are presented in table S2. The additional sulfur in the PS_4^{3-} framework increases the functional capacity and reduces the electrochemical stability of the SSE.”

Comment 14: Comment 9 by Reviewer #2 - Discrepancy Between XANES and XRD Results

The quantification of the LPSCl using XANES is intriguing. However, as stated by reviewer #2 the XANES data in Fig. 4e suggest complete decomposition of LPSCl after discharge, whereas XRD (Fig. 4b) still shows strong LPSCl peaks. The authors attribute this to differences in sulfur loading and the shallow probe depth of XAS but provide insufficient supporting evidence. Given the high LPSCl content, full reduction to LPS is unlikely despite the low areal sulfur loading. To support this claim, the authors should include the corresponding electrochemical data (e.g., voltage profiles, dQ/dV curves) for the cells used in the XRD and XAS experiments in the Supplementary Information. Testing conditions and cell compositions should be clearly specified, and capacity contributions quantified. Additionally, the manuscript should include a concise explanation, with a more detailed discussion in the Supplementary Information, regarding the accuracy and limitations of the XANES quantification.

Response: We thank the reviewer for giving us an opportunity to clarify this discrepancy between the XRD and XANES results. We used linear combination fitting (LCF) of the XANES spectra to identify overlapping redox products between sulfur and LPSCl. More specifically, LCF deconvolutes the relative proportions of different compositions within the sample of interest. This assumes the composite spectra are a linear combination of their individual components. Quantification software like Athena performs non-linear least squares minimization to minimize the error between measured spectra and a linear combination of the reference spectra. The simulated or fitted spectra is a linear combination of the calculated weight corresponding to a particular reference spectrum. Good fits are highly dependent on accurate collection of reference spectra.

The discrepancy between the XRD and XANES results is only applicable for the discharged sulfur sample. The initial XANES analysis resulted in only Li_2S and LPS being reported, while the XRD spectra of a discharged sulfur cathode showed evidence of LPSCI, with minor LPS formation. Given this discrepancy, we re-evaluated the XANES LCF results for this particular sample. The new LCF results were found to deliver a lower R-factor of 0.0017 compared to 0.057, meaning better statistical significance and a better overall fit. A weight distribution of 20.2 wt.% Li_2S , 34.4 wt.% LPS, and 45.3 wt.% LPSCI was determined. The new fit results are shown below.

The new fitting results resolve the discrepancy between the XRD and XANES. The new fit does not significantly alter our conclusion of the SSE redox products and provides a more realistic cathode composition after discharge. If anything, it better supports our hypothesis of LPS to be formed as a redox intermediate with most of the SSE still existing as LPSCI. This helps retain the cathode composite's ionic conductivity.

The new results have been added to the manuscript on line 384 with the new results updated in Fig. 4e. *“The LCF results for the sulfur cathodes are shown in Fig. 4e. After the initial discharge, 20.2 wt.% of Li_2S is estimated. This supports the high conversion efficiency of the cathode architecture. LPSCI also partially decomposes to LPS, as observed for the LPSCI system. A higher amount of LPS is also present than expected but is likely from the $\text{Li}_3\text{PS}_{4+n}$ phase from reacted sulfur.”*

Fig. 4 | Probing electrochemical reversibility. a, Nyquist plots of *in-situ* EIS measurements of the sulfur cathode composite during the first formation cycle. b, XRD and (c) TGA of high loading (7.6 mg cm^{-2}) sulfur cathodes at the pristine, discharged, and charged states. Quantified sulfur masses are listed in *table S4*. XANES fitting results as weight percentages of products at each state of charge for (d) LPSCI/C (e) sulfur/LPSCI/C, and (f) Li_2S /LPSCI/C composites. Error bars represent the uncertainty of the quantified masses. The electrochemical performance of cells used for XRD and XANES are presented in **Fig. S20** and **S21**.

To address the quantification limitations, we also provided the electrochemical data for the cells used for XANES and XRD. The cell configurations have been included within the new Supplementary figures. All cells used for each system (sulfur, Li_2S , LPSCI) deliver comparable electrochemical performance between the different discharged states providing confidence in the XANES quantification trends. The cells used for XRD were fabricated with a higher sulfur loading and areal capacity than for those prepared for XANES. This was done to ensure there was enough sample to conduct air-tight XRD, and ultimately enough sample for phase identification. The high loading sulfur cells deliver reasonable utilization at a very high areal capacity ($\sim 10 \text{ mAh cm}^{-2}$) as shown below in Fig. S20. Therefore, the formed conversion and redox products should be comparable to the lower sulfur loading system. The phases identified with XRD are a bulk analysis of the entire S composite while the XANES is a more local/interfacial analysis on near the cathode surface.

Fig. S20. Electrochemical performance of sulfur cathodes used for XRD analysis.

Fig. S21. Electrochemical performance of cells with a) LPSCI, b) Sulfur, and c) Li_2S cathodes used for XANES quantification.

Table S5. Sulfur K-edge XAS linear combination fitting results.

Composite	Sample	Species Wt. %				χ^2	R-factor
		Sulfur	LPSCI	LPS	Li_2S		
LPSCI/C	Pristine	0	100	0	0	0.03044	0.0005281
	Discharge	0	52.1	47.9	0	0.06229	0.0010326
	Charge	11.60	60.9	27.5	0	0.07749	0.0012868
S/LPSCI/C	Pristine	38.37	61.6	0	0	0.24154	0.0041802
	Discharge	0	45.3	34.4	20.2	0.0983	0.0017195
	Charge	32.0	45.1	22.9	0	0.05798	0.0219914
$\text{Li}_2\text{S}/\text{LPSCI}/\text{C}$	Pristine	0	33.2	31.2	35.6	0.04966	0.0008985
	Charge	45.4	37.6	16.99	0	0.09713	0.0407643
	Discharge	0	34.0	33.3	32.7	0.11070	0.0020389

While the new fitting results present a better fit to the discharged sulfur cathode spectra, the limitations of XANES, due to shallow probe depth and challenges with analysis of the entire

composite originally discussed are still valid and should be mentioned. Clarification and discussion of XANES limitations has been added to the manuscript.

“It must be noted that XANES quantification of the entire composite can be challenging due to the shallow probe depth when using tender X-rays. Discussion regarding limitations of XANES quantification is included in the Supplementary Information.”

We have added a more detailed discussion regarding quantification limitations of XANES within the Supplementary Information.

Quantification limitations of XANES at the Sulfur K-edge

XANES was performed in total fluorescence yield (TFY) mode. In this mode, sulfur spectra can distort due to its strong self-absorption effect.¹ X-ray energies for the Sulfur K-edge are low (2keV) as compared to hard X-rays (5-10 keV), resulting in a limited probe depth of 500 nm to several microns. This narrows the detection region to particle surfaces and their interfaces. The beam spot size for the beamline used in this work is 0.25 mm × 0.16 mm, which covers 0.06% of the electrode area. Therefore, collecting information on the entire cathode composite can be challenging. This is why XRD was conducted in parallel.

XANES quantification was performed using linear combination fitting (LCF) to identify and quantify overlapping redox products between sulfur, LPSCl, and Li₂S. LCF deconvolutes the relative proportions of different references within the sample of interest, assuming the composite spectra are a linear combination of individual components. Quantification software such as Athena (used in this work) performs non-linear least squares minimization to minimize the error between measured and fitted spectra. This fitted spectra is a linear combination of the calculated weights, each corresponding to a specific reference spectrum. Weight uncertainties can be introduced from noise and when the reference spectra exhibit similar features (like LPSCl and LPS), where both possess the thiophosphate unit (PS₄³⁻) within their structure. These factors were considered in our analysis.

1. F. Jalilehvand. Sulfur: not a “silent” element anymore. *Chem. Soc. Rev.*, 2006, 35, 1256–1268.

Comment 15: XPS should be removed from the Author Contribution.

Response: XPS has been removed from the Author Contribution section. We appreciate the careful review by the reviewer.

Reviewer #6

I think the authors have adequately addressed the concerns from the previous reviewers for publication in this journal.

Response: We thank the reviewer for their review and satisfaction of our work for publication in Nature Communications.

Reviewer Comments

Note: Reviewer's comments are shown in **black**, and our responses to the comments are shown in **blue**.

Reviewer #4

All of my comments have now been addressed.

Response: We appreciate the reviewer's satisfaction with our work.

Reviewer #5

I would like to thank the authors for their extensive efforts in addressing all the previous questions. I am in favor of its publication. In addition, please correct the typo, "LiCO2", in Fig. 6c and d to "LiCoO2."

Response: We are grateful for the reviewer's feedback and for their support in publishing our work. We have corrected the typo in Fig. 6c and d.